# Distinct receptor binding domain IgG thresholds predict protective host immunity across SARS-CoV-2 variants and time

Grace Kenny [1,17] ✉, Sophie O'Reilly [1,17], Neil Wrigley Kelly[2,3], Riya Negi[1], Colette Gaillard [1], Dana Alalwan[1], Gurvin Saini[1], Tamara Alrawahneh[1], Nathan Francois[1], Matthew Angeliadis[1], Alejandro Abner Garcia Leon[1], Willard Tinago[1], Eoin R. Feeney[1,2], Aoife G. Cotter[1,4], Eoghan de Barra[5,6], Obada Yousif[7], Mary Horgan[8], Peter Doran[9], Jannik Stemler[10,11], Philipp Koehler [10,11], Rebecca Jane Cox [12], Donal O'Shea[2], Ole F. Olesen[13], Alan Landay[14], Andrew E. Hogan [3,15], Jean-Daniel Lelievre[16], Virginie Gautier[1], Oliver A. Cornely [10,11], Patrick W. G. Mallon[1,2], The All Ireland Infectious Diseases Cohort Study* & VACCELERATE Consortium EU-COVAT-1-AGED Part A Study Group*

SARS-CoV-2 neutralising antibodies provide protection against COVID-19. Evidence from early vaccine trials suggested binding antibody thresholds could serve as surrogate markers of neutralising capacity, but whether these thresholds predict sufficient neutralising capacity against variants of concern (VOCs), and whether this is impacted by vaccine or infection history remains unclear. Here we analyse individuals recovered from, vaccinated or with hybrid immunity against SARS-CoV-2. An NT50 ≥ 100 IU confers protection in vaccine trials, however, as VOC induce a reduction in NT50, we use NT50 ≥ 1000 IU as a cut off for WT NT50 that would retain neutralisation against VOC. In unvaccinated convalescent participants, a receptor binding domain (RBD) IgG of 456 BAU/mL predicts an NT50 against WT of 1000 IU with an accuracy of 80% (95% CI 73–86%). This threshold maintains accuracy in determining loss of protective immunity against VOC in two vaccinated cohorts. It predicts an NT50 < 100 IU against Beta with an accuracy of 80% (95%CI 67–89%) in 2 vaccine dose recipients. In booster vaccine recipients with a history of COVID-19 (hybrid immunity), accuracy is 87% (95%CI 77–94%) in determining an NT50 of <100 IU against BA.5. This analysis provides a discrete threshold that could be used in future clinical studies.

Neutralising antibodies are a key component of the adaptive immune response and are used as a correlate of protection in studies of vaccine-preventable diseases including influenza, measles, polio and yellow fever[1]. More recently, accumulating evidence from vaccine trials, and observational and modelling studies[2–4] have established neutralising antibodies as a correlate of protection against COVID-19[5]. While it is unlikely that a defined neutralising threshold will ever perfectly predict protection, three independent modelling studies of clinical trials using different vaccine platforms have demonstrated excellent vaccine efficacy (ranging from 81–91%) associated with a

A full list of affiliations appears at the end of the paper. *Lists of authors and their affiliations appear at the end of the paper. ✉e-mail: grace.kenny1@ucd.ie

vaccine-induced host neutralising capacity, as measured by the plasma dilution at which 50% of SARS-CoV-2 infection was preserved in vitro (NT50), of 100 international units (IU)[6–8]. While these studies also examined the association of binding IgG titres with vaccine efficacy, the emergence of variants that evade the neutralising response limits the applicability of these titres.

Neutralising assays, particularly gold standard live virus assays, are labour intensive, costly, and require skilled staff and high biosafety level requirements, preventing their use in large-scale vaccine trials or roll out for use in routine clinical settings. While neutralising antibodies against SARS-CoV-2 correlate with anti-spike IgG binding antibodies, there is a significant inter-individual variation in this correlation[9], which has limited the implementation of binding antibody assays to estimate an individual's risk of developing severe infection[10]. Consistent with this, reinfection, including severe infection, is well described in seropositive individuals with a history of COVID-19[11,12]. Given the ongoing circulation of SARS-CoV-2, a binding IgG titre that could predict adequate protection against severe COVID-19 is urgently needed, not least to guide the timing of further vaccine doses and to identify those with inadequate immunity (including vaccine failures) in which use of effective but limited COVID-19 therapeutics should be prioritised[13]. The search for a correlate of protection has been complicated both by the emergence of variants of concern (VOCs) which differ in degree of immune evasiveness, and the heterogeneity in population immunity, which is impacted by a variety of factors including SARS-CoV-2 infection history, vaccine platform, number of vaccine doses, time between vaccine doses and time since last vaccine or infection[14,15].

To determine if a binding IgG titre can reliably predict robust host neutralising capacity, this study aims to define relationships between binding IgG and neutralising capacity using a live virus neutralising assay in individuals with immunity from natural infection, vaccination or both, explore thresholds of immunity, associations with clinical factors and cellular immunity, and the impact on these associations of SARS-CoV-2 variants of concern (VOC) that confer immune escape.

## Results

### Participant demographics
We looked at three separate groups for this analysis. The demographics of each group are outlined in Table 1. The first was an unvaccinated, convalescent group of 131 individuals with PCR-confirmed COVID-19 that provided 190 samples for analysis. All participants had acute symptoms in March to May 2020, the first wave of COVID-19 in Ireland, before the introduction of vaccination or detection of VOC[16]. The second group included 55 individuals who had received a two-dose primary vaccine series (predominantly mRNA vaccines), with a sample collected at least 14 days after the second vaccine dose. 23 (42%) of this group were individuals with a history of PCR-confirmed COVID-19. The last group were 70 individuals who had received a primary COVID-19 vaccine course and one booster vaccine and had a history of at least one SARS-CoV-2 infection within the Omicron-dominant period in Ireland.

### Relationship between NT50 and IgG within the unvaccinated group
We first looked to establish a threshold that predicted an NT50 of <1000 IU against WT-B in the unvaccinated group. All antigenic targets were significantly positively correlated with NT50, but RBD had the strongest correlation (rho 0.81, $p < 0.001$, Fig. 1a) although S1 was also highly correlated (rho = 0.80 $p < 0.001$). Correlations between RBD IgG and NT50 further strengthened when the analysis was restricted to those >30 days post symptom onset when the contribution of unmeasured IgM antibodies to neutralisation is likely to have waned (rho 0.85 for RBD and 0.84 for S1, both $p < 0.001$) (Supplementary Table 1).

Linear regression estimated the titres corresponding to a WT-B NT50 of 1000 IU threshold were 574 BAU/mL for RBD, 896 BAU/ml for S1, 303 BAU/mL for S2 and 439 BAU/mL for nucleocapsid. Using data from the whole unvaccinated group, the RBD threshold had the overall best sensitivity and specificity to predict a WT-B NT50 of <1000 IU (sensitivity 83% (95%CI 76–89%), specificity 73% (95%CI 56–87%)), which improved when the analysis was restricted to samples taken in the convalescent period, >30 days from symptom onset ($n = 148$), giving a sensitivity of 82% (95% CI 74–88%) and specificity of 80% (95% CI 56–94%), respectively (Supplementary Table 2).

ROC curves were constructed to demonstrate the performance of each antibody measurement in predicting WT-B NT50 < 1000 IU. AUC for RBD was 0.86 (95% CI 0.79–0.93, Fig. 2a), for S1 it was 0.85 (0.78–0.91), for S2 it was 0.75 (95% CI 0.67–0.84) and for nucleocapsid 0.84 (95% CI 0.76–0.91). Once again these estimates improved when the analysis was restricted to the convalescent setting (RBD AUC 0.9 (95% CI 0.85–0.95), S1 0.85 (0.78–0.91), S2 0.8 (95% CI 0.72–0.89) and 0.84 (95% CI 0.74–0.93) for nucleocapsid.

The Youden index for RBD at 456 BAU/mL improved specificity compared to the linear regression threshold with a sensitivity of 77% (95% CI 69–84%) and specificity of 100% (95% CI 82–100%) in samples >30 days post symptom onset, with an overall accuracy of 80% (95% CI 73–86%). Although the sensitivity was poorer, the median NT50 of those samples ($n = 29$) with an RBD > 456 BAU/mL but an NT50 of <1000 IU was 589 (IQR 415–749) IU, and all had an NT50 > 100 IU reinforcing the robustness of a discrete RBD threshold to reliably predict protective underlying neutralising capacity.

### Comparison of T-cell responses to WT NT50 in unvaccinated participants
To explore the relationship between WT-B NT50 and cellular immunity, we assessed spike-specific CD4+ and CD8+ responses in a subset of 33 unvaccinated individuals. We assessed the frequency of polyfunctional (expressing both IFNγ + TNFα+) spike-specific CD4+ and CD8 + T-cells, IFNγ + spike-specific CD4+ and CD8 + T cells and CD8 + TNFα + CD69 + cells. There was no correlation with NT50 for any parameter tested (all $p > 0.05$, Supplementary Table 3). Additionally, looking at 12 individuals with an RBD > 456 BAU/mL and 21 individuals with an RBD < 456 BAU/mL, there were no significant differences in the proportion of spike-specific cells between the two groups by any parameter tested ($p > 0.05$, Supplementary Table 4). Only two individuals had no detectable spike-specific T-cell response by any of the parameters described, both of whom were in the low RBD group. As the T-cell response did not correlate with NT50, we focused on IgG thresholds for further analysis.

### Impact of beta variant on neutralising capacity in unvaccinated participants
To test our assumption that a discrete RBD of 456 BAU/mL titre would predict robust immunity that retains protective neutralising capacity against an immune escape VOC, we first examined NT50 against the Beta variant in a subset of 36 of the unvaccinated individuals with the highest RBD titres. As we had observed a weaker correlation between NT50 and RBD IgG titre in acute samples, potentially due to neutralisation from unmeasured IgM antibodies, we selected individuals that were at least 60 days post symptom onset to ensure this reflected the truly convalescent period. Samples within this subset were taken with a median (IQR) of 137 (100–184) days from the acute onset of infection, had a median (IQR) RBD of 684 BAU/mL (525–939), a WT-B.1.177.18 NT50 of 850 IU (372–544) and displayed an abrogated median Beta NT50 of 298 IU (138–736). Specificity of an RBD titre of 456 BAU/mL to rule out a Beta NT50 of <100 IU was excellent at 90% (73–97%), although sensitivity was lower at 57% (95%CI 18–90%), with significant uncertainty around this estimate due to low numbers of those with an NT50 < 100 against Beta ($n = 7$). However, the overall

**Table 1 | Participant demographics**

| | Unvaccinated convalescent N = 131 | Two-dose vaccine N = 55 | Booster vaccine hybrid N = 70 |
|---|---|---|---|
| Age (median (IQR)) | 51 (41–64) | 46 (35–55) | 42 (33–51) |
| Female sex (n(%)) | 67 (51) | 32 (57) | 41 (57) |
| BMI (median (IQR)) | 29 (23–33) | 24 (22–29) | 26 (23–30) |
| History of COVID-19 (n(%)) | 131 (100) | 25 (45) | 70 (100) |
| WHO disease severity[a] **(n(%))** | | | |
| Mild | 92 (70) | 21 (84) | 62 (88) |
| Moderate | 23 (18) | 3 (12) | 2 (3) |
| Severe | 5 (4) | 1 (4) | 2 (3) |
| Critical | 10 (7) | 0 (0) | 4 (6) |
| Days from most recent infection (median (IQR)) | 99 (35–179) | 197 (141–358) | 151 (104–217) |
| Primary vaccine series | NA | | |
| BNT162b2 | | 48 (87) | 55 (78) |
| mRNA-1273 | | 6 (11) | 5 (7) |
| Other | | 1 (2) | 10 (14) |
| Booster vaccine | NA | NA | |
| BNT162b2 | | | 66 (94) |
| mRNA-1273 | | | 4 (5) |
| Days from most recent vaccine dose (median (IQR)) | NA | 55 (36–104) | 252 (191–309) |
| RBD (median (IQR)) | 246 (71–662) | 3132 (1278–8412) | 4084 (1617–8399) |
| S1 (median (IQR)) | 307 (85–770) | 4277 (1397–16787) | 39241(13073–80223) |
| S2 (median (IQR)) | 165 (78–370) | 135 (19–653) | 699 (311–80223) |
| Nucleocapsid (median (IQR)) | 65 (18–378) | 25 (11–101) | 33 (20–114) |
| WT NT50 (median (IQR)) | 165 (49–710) | 1030 (286–4391) | 3303 (1372–7457) |
| Beta NT50 (median (IQR)) | 298 | 408 (107–1497) | NA |
| Omicron NT50 (median (IQR)) | NA | NA | 427 (234–914) |

In the unvaccinated convalescent group, 39 individuals contributed 2 samples, 7 contributed 3 samples, and 2 contributed 4 samples.
*BMI* body mass index, *IQR* interquartile range, *NA* not assessed.
[a]Graded as per the World Health Organisation (WHO) severity scale[48].

accuracy of the RBD threshold of 456 BAU/mL to predict Beta NT50 100 IU was 83% (95% CI 67–94%).

**Performance of RBD thresholds in participants following primary vaccine series**
We next looked within a group (*n* = 55) that had received a 2-dose primary vaccine series and were a median of 55 (IQR 36–104) days from the second vaccine dose. Overall RBD and NT50 were higher within this group (median (IQR) RBD 3341 (1274–8376) BAU/mL, WT-B.1.177.18 NT50 1030 (285–4390) IU, Beta NT50 408 (107–1497) IU, Fig. 3), and only 4 individuals (7%) had an RBD of <456 BAU/mL. RBD again correlated strongly with both WT-B.1.177.18 NT50 (spearman rho 0.76, *p* < 0.001, Fig. 1b), and to a lesser extent Beta NT50 (rho 0.64 *p* < 0.001, Fig. 1c). Those with a history of previous COVID-19 had significantly higher RBD and NT50 titres (median (IQR) RBD 7052 BAU/mL (3867–20409) vs 1348 BAU/mL (668–3250), WT-B.1.177.18 NT50 4308 IU (1055–5670) vs 451 IU (90–1100), Beta NT50 1432 IU (379–3513) vs 238 IU (56–548) in those with vs without a history of infection respectively, all *p* < 0.001. Sensitivity of an RBD threshold of 456 BAU/mL was poorer in this two-dose vaccination population at 23% (95%CI 5–53%), likely due to the small number of observations, although specificity remained high at 98% (87–99%). Both positive predictive value (PPV, i.e., the likelihood of having sub-protective immunity with an RBD < 456 BAU/mL) and negative predictive value (NPV, i.e., the likelihood of having protective levels of immunity with an RBD > 456 BAU/mL) were 80% (95% CI 28–99%) and 80% (95%CI 68–91%) respectively, and overall accuracy remained high at 80% (95% CI 67–89%). We observed that the vast majority (92%) of the vaccinated

population that had an NT50 < 100 IU against the Beta variant were those without a prior history of COVID-19.

**Performance of RBD thresholds in a population with hybrid immunity**
We next explored the utility of these thresholds in a group with hybrid immunity, more reflective of contemporary population immunity. This group had received primary vaccination, and one booster vaccine and also reported a history of at least one confirmed infection within the Omicron VOC period. We assessed NT50 against WT-B.1.177.18 and the BA.5 subvariant of the Omicron lineage. Median (IQR) RBD IgG was 4084 BAU/mL (1617–8399), which was not significantly different to the 2-dose vaccine group (*p* = 0.86), but median WT-B.1.177.18 NT50 was significantly higher at 3303 IU (IQR 1372–7457) compared to 1030 IU (IQR 286–4391) in the 2-dose vaccine group (*p* < 0.001)(Fig. 3). Median (IQR) BA.5 NT50 was 427 IU (234–914). In this group, RBD significantly correlated with both WT-B.1.177.18 NT50 (rho 0.76, *p* < 0.001, Fig. 1d) and BA.5 NT50, although the strength of the correlation was abrogated (rho 0.53, *p* < 0.001, Fig. 1e). Sensitivity and specificity of the RBD IgG threshold of 456 BAU/mL were similar to the 2 dose vaccine population at 33% (10–65%) and 98% (91–99%) respectively, again reflecting small numbers of observations while PPV and NPV were better at 80% (28–99%) and 88% (77–94%) respectively, and overall accuracy of the RBD threshold of 456 BAU/mL to predict BA.5 NT50 of 100 IU remained high 87% (95% CI 77–94%).
As overall RBD IgG levels were higher in both vaccinated groups compared to the convalescent group, we next explored if determining an RBD threshold that predicted a WT NT50 of <1000 IU within these

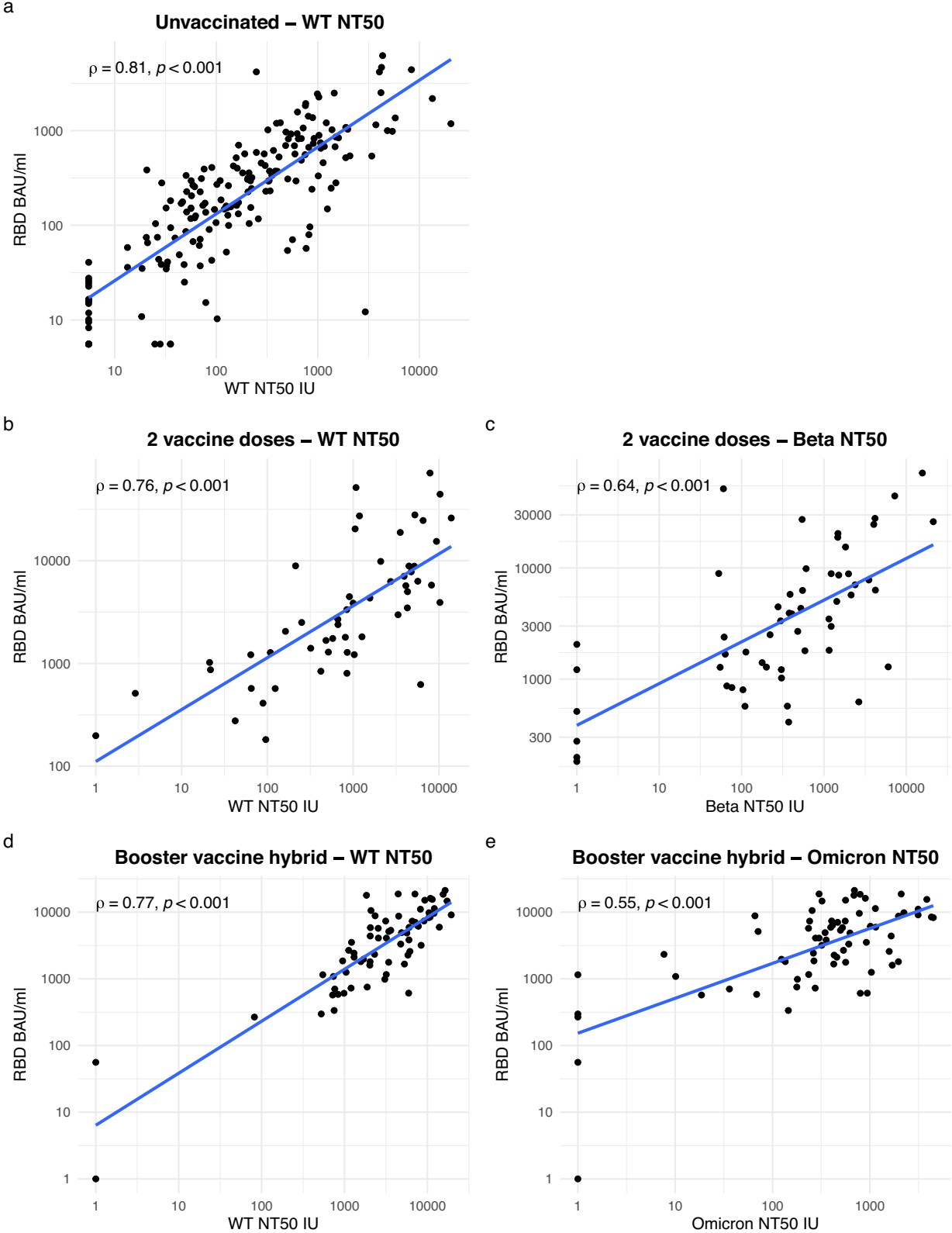

**Fig. 1 | Correlation between RBD and NT50.** Two-tailed Spearman correlation between RBD and NT50. **a** Unvaccinated convalescent participants and WT-B NT50 (exact *p*-value < 2.2e-16). **b** Participants who had received only a primary vaccine series and WT-B.1.177.18 NT50 (exact *p*-value < 2.2e-16). **c** Participants who had received only a primary vaccine series and Beta NT50 (exact *p*-value = 1.381e-07). **d** Participants who had received one booster vaccine dose and had a history of infection and WT-B.1.177.18 NT50 (exact *p*-value = 3.99e-15). **e** Participants who had received one booster vaccine dose and had a history of infection and Omicron NT50 (exact *p*-value = 1.042e-06). Source data are provided as a source data file.

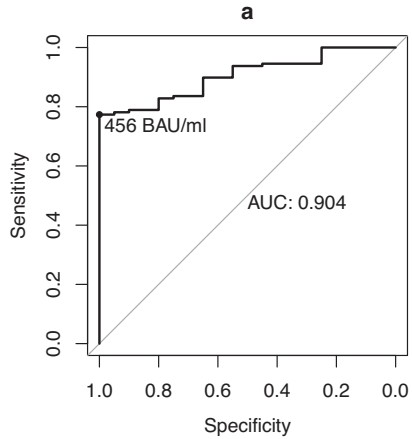
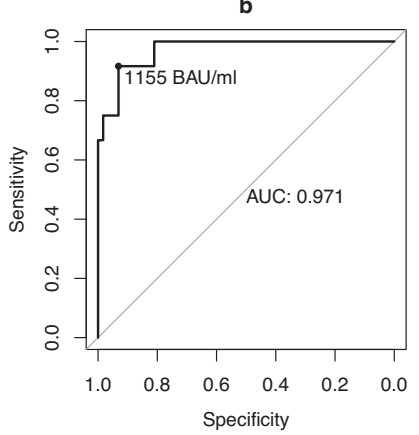

**Fig. 2 | Receiver operating characteristic curves. a** Receiver operating characteristic (ROC) curve demonstrating the performance of RBD in discriminating a wild type (WT-B) NT50 < 1000 IU in unvaccinated participants. **b** ROC curve demonstrating the performance of RBD in discriminating a wild type (WT-B.1.177.18) NT50 < 1000 IU in participants with one booster vaccine and a history of infection within the Omicron period. Source data are provided as a Source Data file.

populations could better predict retention of NT50 of 100 IU against VOC. Repeating the ROC analysis within the hybrid immunity group, the AUC of the ROC curve to predict a WT-B.1.177.18 NT50 of <1000 IU was excellent at 0.97 (95% CI 0.93–1)(Fig. 2b). The Youden index gave an RBD of 1155 BAU/mL, which was associated with a sensitivity of 92% (95% CI 61–99%) and specificity of 93% (95% CI 83–98%) to predict a WT-B.1.177.18 NT50 of <1000 IU. Using this revised RBD threshold, against BA.5, sensitivity and specificity to predict an NT50 < 100 IU were 75% (95% CI 43–94%) and 90% (79–96%) respectively, with a corresponding PPV of 60% (32–84%) and NPV of 94% (84–99%), and overall accuracy of 87% (77–94%).

### Clinical factors associated with neutralising capacity against Omicron

Given the abrogated correlation between RBD and BA.5 NT50 compared to WT-B.1.177.18 NT50, we explored whether clinical factors could impact the association between RBD and both WT-B.1.177.18 and BA.5 NT50. In univariate analysis, there was no association with age, sex, BMI or days since the most recent booster (all $p > 0.05$, Table 2). Only a longer time since the most recent infection was associated with a decrease in both WT and BA.5 NT50 (−15 unit decrease in WT NT50 per day longer from infection, $p = 0.008$ and −2.9 unit decrease in BA.5 NT50 per day further from infection, $p = 0.02$). In multivariable linear regression, RBD remained significantly associated with WT-B.1.177.18 NT50 and BA.5 NT50, while days from symptom onset remained significantly associated with BA.5 NT50 ($p = 0.04$) but not WT-B.1.177.18 NT50 ($p = 0.06$, Table 2).

### Validation in the VACCELERATE EU-COVAT-1 AGED trial

To validate our findings, we next explored the performance of the 456 BAU/mL RBD IgG threshold in predicting the retention of neutralising capacity against an immune escape variant in an independent cohort. We utilised samples from Part A of the EU-COVAT-1 AGED trial, which included individuals aged ≥75 years sampled prior to and 14 days after the third dose vaccination. We selected this cohort as older individuals were under-represented in the other groups analysed, and are of particular interest given the greater risk of severe disease with increasing age. The EU-COVAT-1 AGED trial used a commercially available serologic assay to quantify anti-RBD IgG and the analysis included 100 samples from 50 individuals, the characteristics of which have been described previously[17]. In this group, RBD IgG remained significantly correlated with WT-B.1.177.18 NT50 (rho 0.9, $p < 0.0001$) and Beta NT50 (rho 0.86, $p < 0.0001$)(Supplemental Fig. 3). An RBD IgG of 456 BAU/mL predicted a WT-B.1.177.18 NT50 of <1000 IU with a sensitivity of 59%

(48–70%), a specificity of 100% (83–100%) and an overall accuracy of 68% (58–77%). Performance of this threshold in predicting a Beta NT50 of <100 IU remained excellent, with a sensitivity of 92% (80–98%) a specificity of 96% (86–99%) and an accuracy of 94% (87–98%).

## Discussion

In this study, we established a threshold of RBD of 456 BAU/mL that accurately predicted a clinically significant host viral neutralising capacity against WT SARS-CoV-2 that was retained against two immune escape variants. This threshold remains accurate when tested in diverse populations with heterogeneous host immunity from different stages of the COVID-19 pandemic, and when measured using a commercially available assay in an independent cohort. While the performance of thresholds varied across groups, at each stage, they offered performance characteristics that could be used to support clinical decision-making. This demonstrates the feasibility of using a binding IgG threshold as a surrogate for neutralising capacity, offering the potential for the use of a simplified laboratory assay to determine host immunity to SARS-CoV-2.

Despite the wealth of evidence linking neutralising antibodies to protection against disease, the FDA currently recommends against using antibody testing to guide decisions around vaccination timing, use of prophylactic monoclonal antibody therapy and other therapies[10]. This is primarily due to a lack of research evaluating the use of available antibody tests for these purposes. As SARS-CoV-2 era progresses, decisions on further vaccine doses or other therapies will likely become tailored to individual circumstances, including preference, risk factor profile, adverse reactions to vaccines and, importantly, underlying host susceptibility to symptomatic COVID-19. This study provides a basis for the use of a threshold of RBD titres of 456 BAU/mL to provide an accurate indication of underlying host immunity to SARS-CoV-2 and to identify individuals with suboptimal immunity.

While no threshold will ever perform perfectly at predicting protection against symptomatic COVID-19, many screening tests used in clinical practice, such as tumour markers, have false positive and negative rates, and yet still provide useful information for patient care[18]. Quantitative antibody thresholds that reflect clinically relevant host neutralising capacity, the gold standard surrogate of underlying host immunity, could be used to assist clinical decision-making, including identification of vulnerable individuals with suboptimal immunity for vaccination as well as directing appropriate therapeutics such as antivirals, for those with suboptimal immunity who become infected with SARS-CoV-2. Further clinical studies will be required to explore the utility of these thresholds in guiding these clinical decisions.

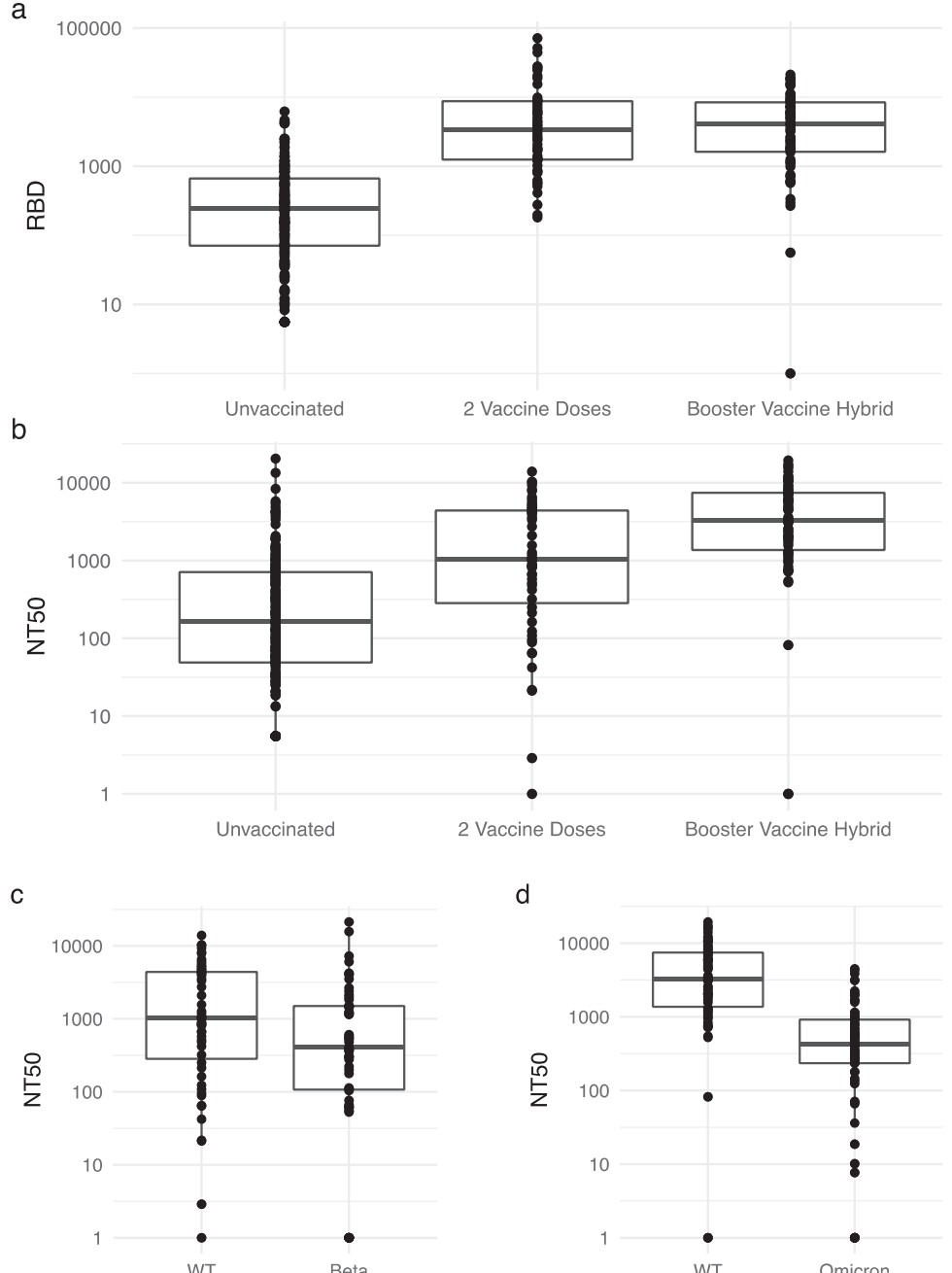

**Fig. 3 | RBD and NT50 levels between groups. a** Boxplots showing RBD levels in each group of participants (unvaccinated $n = 190$, 2 vaccine dose $n = 55$, booster vaccine hybrid $n = 70$ biologically independent samples). **b** Boxplots showing NT50 against WT in each group of participants (unvaccinated $n = 190$, 2 vaccine dose $n = 55$, booster vaccine hybrid $n = 70$ biologically independent samples). **c** Boxplots showing NT50 against WT-B.1.177.18 and Beta in the two-dose vaccine dose group ($n = 55$ biologically independent samples). **d** Boxplots showing NT50 against WT-B.1.177.18 and Omicron in the booster vaccine hybrid group ($n = 70$ biologically independent samples. The centre of the boxplots represents the median, the bounds of the box represent the interquartile range, and the whiskers represent the minimum and maximum values no more than 1.5 times the interquartile range. Where the minima and maxima lie outside the whiskers they are represented as dots. Source data are provided as a Source Data file.

Previous studies have demonstrated that the number and type of antigenic exposure affect the breadth of the humoral response[19]. In line with this, we found infection history was an important factor that impacted neutralising potency. The majority of the vaccinated groups that failed to retain protective neutralising capacity against the Beta variant were infection naïve, and time from the most recent infection was associated with neutralising titres in the booster vaccine group. While this suggests that there may be a lower overall risk of individuals having suboptimal neutralising capacity in current populations with high levels of vaccination and infection, further studies are needed, particularly in more vulnerable groups, such as those on immunomodulating therapies that affect B cell function, to determine whether RBD titres reflect sufficient neutralising capacity in all groups.

Although the correlation between viral neutralising capacity (NT50) and anti-spike targeted antibodies has previously been reported[20,21], the identification of a discrete antibody threshold that predicts a clinically relevant underlying viral neutralising capacity against both wild type and immune escape variants of SARS-CoV-2 has not been well defined.

**Table 2 | Association between clinical variables and NT50 in booster vaccine cohort**

| | WT NT50 Change (IU) (95% CI) | p-value | WT NT50 Change (IU) (95% CI) | p-value |
|---|---|---|---|---|
| Univariable analysis | | | Multivariable analysis | |
| Age (years) | 51.04 (−24.24–126.33) | 0.18 | | |
| Male sex | −295.3 (−2549.63–1959.1) | 0.79 | | |
| BMI | −21.47 (−69.55–26.6) | 0.37 | | |
| Days since booster | −11.75 (−24.86–1.41) | 0.08 | | |
| Days since most recent infection | −15.78 (−27.32 to −4.25) | 0.008 | −2.37 (−4.8–0.069) | 0.06 |
| RBD | 0.06 (0.02–0.12) | 0.002 | 0.06 (0.02–0.099) | 0.006 |
| | Omicron NT50 Change (IU) (95% CI) | p-value | Omicron NT50 Change (IU) (95% CI) | p-value |
| Univariable analysis | | | Multivariable analysis | |
| Age (years) | 3.67 (−12.77–20.12) | 0.67 | | |
| Male sex | 62.69 (−424.17–549.54) | 0.79 | | |
| BMI | −2.26 (−11.75–7.22) | 0.63 | | |
| Days since booster | 1.77 (−1.9–4.64) | 0.22 | | |
| Days since most recent infection | −2.91 (−5.44 to −0.39) | 0.02 | −2.53 (−4.94 to −0.12) | 0.04 |
| RBD | 0.03 (0.01–0.05) | 0.003 | 0.03 (0.008–0.04) | 0.004 |

Results from univariable and multivariable linear regression models exploring the impact of clinical variables on the association between RBD and NT50.
*IU* International Units, *BMI* body mass index, *RBD* receptor binding domain.

Research in this area has focused on identifying high-titre convalescent plasma donors using commercially available serologic tests. The tests evaluated generally perform favourably at predicting the presence of any neutralising capacity, but lose accuracy when evaluating higher neutralising thresholds[22–25], or focus only on WT neutralising capacity[26–28]. One other study has examined IgG thresholds to predict neutralising titres against VOCs[29], but examined only convalescent and those vaccinated with either one or two doses of BNT162b2 or ChADOx1, and the assays were not standardised to international units. We used a robust, well-characterised, quantitative multiplex antibody assay[30] and a quantitative, live virus neutralising assay, both reporting in international standardised units, enabling a comprehensive assessment of the relationship between serologic responses and neutralising capacity against wild type and immune escape SARS-CoV-2 variants which can therefore be compared and validated to results from other studies.

Independently, other studies support the rationale for a threshold of immunity similar to the target identified here, with an RBD of 506 BAU/mL estimated to provide a vaccine efficacy of 80% against symptomatic COVID-19 in one study[3] during which the alpha variant was prominent (n = 1318). Another study from Israel (n = 1461) showed the probability of becoming infected after exposure was 11% with an RBD > 500 BAU/mL compared to 38% in those with undetectable RBD during the delta dominant period[31]. While an RBD threshold that predicts protection against Omicron infection has not been determined, the ongoing association between anti-spike titres and protection has been demonstrated[32].

Our observation that a higher RBD titre of 1155 BAU/mL may confer improved prediction of retention of NT50 > 100 IU is in keeping with the greater immune escape displayed by VOC[14]. However, a serologic test only reflects circulating antibodies at the time of testing and a single RBD level is unlikely to predict the variable individual dynamics of antibody titres over time[33]. Laboratory-based quantitative antibody testing with a short turnaround time or emerging point-of-care antibody testing may allow for more frequent monitoring of vulnerable individuals at risk of poor immune responses and/or severe COVID-19, and improve the potential for appropriate, targeted timing of prophylactic and therapeutic interventions, such as booster vaccinations or use of antivirals in those with presenting early in infection.

This study has limitations. Although we looked at four groups, each with different types of immunity to SARS-CoV-2, we did not include those who had received bivalent boosters, and we did not examine neutralisation against later emerging SARS-CoV-2 variants (BA.5) in the earlier convalescent, two-dose vaccine or validation cohorts. We also focused on an RBD assay derived from WT RBD, although we found no meaningful difference when we validated our findings using an Omicron-specific RBD. We provide only in vitro data on host viral neutralisation, and although we know the infection history of the participants of this study, our study is cross-sectional and data on subsequent infection or severity was not available. The cross-sectional nature also precluded examination of the dynamic performance of RBD thresholds over time post-infection or vaccination. We used NT50 thresholds derived from published clinical trials that were associated with protection against symptomatic infection, and thresholds associated with protection against hospitalisation, severe disease and death may be more clinically meaningful endpoints, although these are usually lower than thresholds that protect against infection. Although we used a live virus neutralisation assay, regarded as the gold standard, the neutralising thresholds chosen were derived from prior studies using pseudovirus assays, and while normalised to an international standard to allow comparison, we did not directly assess the correlation between these assays. We assumed that the 100 IU threshold would remain protective against variants as suggested by modelling studies[34,35], but this has not been explicitly tested. Future variants may display even greater immune escape and require a higher threshold to confer protection unpredictable nature, but the approach outlined here can be used to estimate any change in RBD IgG titre that may be required to protect against future novel SARS-CoV-2 variants. Despite these limitations, we used live virus assay to examine neutralising capacity against three different variants, evaluated both humoral and cellular immunity and undertook an analytic approach that can be validated in future, prospective studies in diverse populations.

In summary, we describe a discrete RBD threshold of 456 BAU/mL that corresponds to robust underlying protective immunity, measured by neutralising capacity against both wild-type and immune escape variants of SARS-CoV-2. Further studies are needed to determine how best to implement these thresholds in clinical practice.

## Methods
### Study design and participants
The All Ireland Infectious Diseases Cohort (AIID) study is a prospective, multi-centre observational cohort study that recruits individuals attending clinical services for issues relating to infectious diseases from nine clinical centres in Ireland. Participants provided written,

informed consent for the collection of demographic and clinical data and the collection of blood samples for biobanking, from which plasma was stored at −80 °C and peripheral blood mononuclear cells (PBMCs) in liquid nitrogen until analysis. Participant data were collected and managed using a REDCap (v11.1.8) database hosted at University College Dublin[36] Individuals with PCR-confirmed COVID-19 and/or receipt of COVID vaccination were included in this study. The AIID Study and these analyses were approved by the St Vincent's Hospital group Research Ethics Committee and the National Research Ethics Committee for COVID-19 in Ireland.

### Validation cohort

EU-COVAT-1 AGED is a multinational, phase 2, randomised clinical trial examining the immunogenicity, reactogenicity and safety of a third vaccine dose in adults ≥75 years of age, conducted within the VAC-CELERATE network (NCT 05160766). The trial design and results of EU-COVAT-1 have been described in detail previously[17,37]. Briefly, participants ≥75 years of age who had received homologous ChAdOx-1-S, BNT162b2 or mRNA-1273 priming vaccine regimens and had no SARS-CoV-2 infection within the preceding 3 months were randomised to receive a third dose of either BNT162b2 or mRNA-1273 in a 1:1 ratio. Blood was drawn on the day of vaccination and at day 14 for immunogenicity analysis. EU-COVAT-1 was approved by the Ethics Committee of the Faculty of Medicine, University of Cologne, Germany. We used plasma samples derived from baseline and day 14 visits from participants of part A of this trial for analysis.

### CEPHR COVID-19 Serologic assay

IgG against the receptor binding domain (RBD) of WT and Omicron, spike subunits 1 and 2 (S1, S2) and nucleocapsid (N) proteins were measured using the CEPHR COVID19 serologic assay, which has been described in detail elsewhere[30]. Briefly, SARS-CoV-2 RBD, S1, S2 and N (Sino Biological, Inc), diluted in ChonBlock ELISA buffer (CB) (Chondrex Inc, Redmond, WA, USA), were coupled with individual MSD "linkers" Meso Scale Diagnostics, LLC (MSD, Rockville, MD). All linkers were then combined to make a coating solution which was added to each well of MSD 96 well U-PLEX plates and incubated. Plates were then washed in phosphate-buffered saline (PBS) and 0.05% tween (Bio Sciences Ltd, Ireland). Serial dilutions of RBD, S1, S2 and N antibodies (Sino biological, Inc) at a concentration of 1.25 μg/mL, 5 μg/mL, 4 μg/mL and 3 μg/mL, respectively, in CB were used to make a 7-point standard curve. Plasma diluted in CB and standard were added to the wells, and plates were incubated and then washed. MSD SULFO-TAG-labelled goat anti-human IgG secondary antibody was added to the plasma wells, MSD SULFO-TAG-labelled goat anti-mouse and anti-rabbit IgG was added to the standard curve wells all at a 1 μg/mL dilution, incubated, washed and MSD GOLD read buffer B added. Plates were then analysed with a MESO QuickPlex SQ 120 instrument (MSD, Rockville, MD, USA). The operational performance of this assay, and harmonisation of the assay output to the first World Health Organisation (WHO) international standard for anti-SARS-CoV-2 immunoglobulin (National Institute for Biological Standards and Control (NIBSC) code 20/136, Hertfordshire, UK, reported as international units (IU)/mL) has been outlined in detail elsewhere[30].

### Meso Scale Diagnostics V-PLEX SARS-CoV-2 Serologic Assay

Anti-RBD IgG was measured in the EU-COVAT-1 AGED cohort using the commercially available MSD V-PLEX SARS-CoV-2 Panel 2 Kit (MSD, Rockville, MD, USA), as per the manufacturer's instructions. Briefly, plates were blocked with MSD Blocker A for 30 min then washed with MSD wash buffer. Plasma samples, standards and control samples were diluted in MSD diluent 100, added to the plate and incubated for 2 h. Plates were then washed with MSD wash buffer and the detection antibody diluted in MSD diluent 100 to a concentration of 1 μg/mL was added to the plate and incubated for 1 h. Plates were washed with MSD

wash buffer, MSD GOLD Read Buffer B was added, and plates were analysed with a MESO QuickPlex SQ 120 instrument (MSD, Rockville, MD, USA). Results were normalised to the WHO international standard using the conversion factor provided by the manufacturer.

### Flow cytometry-based micro-neutralisation assay

SARS-CoV-2 isolates used in this study were WT-B (Pango Lineage B (WT-B) clinical isolate 2019-nCoV/Italy-INMI1 from the European Virus Archive goes Global (EVAg), Spallanzani Institute, Rome[38], and WT with D614G substitution (CEPHR_IE_B.177.18B.1.177.18_1220, GenBank accession ON350866), Beta (SARS-CoV-2/human/IRL/AIIDV1752/2021, GenBank: ON350868.1) and Omicron-BA.5 (Pango lineage BA.5, GenBank accession OP508004) each isolated from SARS-CoV-2 positive nasopharyngeal swabs from the AIID cohort. Viral neutralisation was measured on Vero E6 cells (VERO C1008, Vero 76, clone E6, Vero E6, ATCC, Manassas, VA, USA) or Vero E6/TMPRSS2 cells (#100978) obtained from the Centre For AIDS Reagents (CFAR) at the National Institute for Biological Standards and Control (NIBSC)[39,40] using a flow-cytometry-based micro-neutralisation assay as described[41]. Briefly, plasma was serially diluted with an 8-point, half-log dilution from 1:20 to 1:62,927 then co-incubated with SARS-CoV-2 at a 1:1 ratio at 37 °C, 5% $CO_2$ for 1-hr. The virus-plasma mixture was added to confluent cells. Infection was measured at 18-hrs post-infection via intracellular SARS-CoV-2 Nucleoprotein (NP) staining (Invitrogen SARS/SARS-CoV-2 Nucleocapsid Monoclonal Antibody (E16C), 1:100 dilution, Santa Cruz Biotechnology goat anti-mouse IgG2b-FITC, 1:500 dilution) detected by flow-cytometry (CytoFlex S, Beckman Coulter). The gating strategy is shown in Supplementary Fig. 1. The plasma dilution resulting in a 50% reduction in infection (NT50) was determined using logistical regression (4-paramater, variable slope) with GraphPad Prism (Version 9.3.1). NT50 values were converted to International Units (IU) using the First WHO International Reference Panel for anti-SARS-CoV-2 Immunoglobulin NIBSC code: 20/268. Briefly, NT50 values obtained with WHO panel of pooled convalescent samples (low, mid or high anti-SARS-CoV-2 Spike IgG titres) on Vero E6 or Vero E6/TMPRSS2 cells were plotted against the IU provided for each WHO standard. We used the equation of the line to apply the conversion to the plasma samples tested.

### Antigen-specific T-cell response

Spike-specific CD4 and CD8 T-cell response was measured as previously described[42]. Biobanked PBMCs were thawed in batches and re-suspended in cRPMI. Cells were incubated in the presence of SARS-CoV-2 spike peptide pools (Miltenyi PepTivator®SARS-CoV-2) at a concentration of 1 μg/mL and $5-10 \times 10^6$ cells/mL. After 4 h, protein transport inhibitor of brefeldin A and monensin (eBioscience) was added to each well and samples were incubated at 37 °C for a further 12 h.

Cells were then stained for cell surface markers (CD3 VioGreen 1:100 dilution, CD4 PE-Vio 770 1:100 dilution, CD8 PerCP-Vio700 1:100 dilution, CD69 VioBlue 1:100 dilution, (Miltenyi Biotec), PD-1 BV711 1:40 dilution (BioLegend) and "fixed", permeabilised and stained for intracellular cytokines (IFNγ Alexa Flour 488 1:40 dilution (Biolegend) and TNFα APC 1:100 dilution (Miltenyi Biotec)). Multicolour flow cytometry was performed using Attune™ NxT Flow Cytometer. The gating strategy is shown in Supplementary Fig. 2.

### Statistical analysis

Quantitative and qualitative variables were summarised with median and interquartile range (IQR) and number and percent, respectively. Spearman correlation was used to assess relationships between IgG and NT50, and both IgG and NT50 and spike-specific T-cell responses.

Linear regression analyses and receiver operating characteristic (ROC) curves were used to determine the IgG threshold that best predicted an NT50 of 1000 IU. In ROC curve analysis we selected the

point that maximises sensitivity and specificity (Youden Index)[43]. Two modelling studies, based on the mRNA-1273[6] and NVX-CoV2373[7] COVID-19 vaccine efficacy trials, have demonstrated excellent vaccine efficacy against symptomatic infection with a post-vaccine NT50 of 100 IU. In the mRNA-1273 trial, a higher NT50 of 1000 IU was associated with a relatively marginal increment in protection with a vaccine efficacy increasing from 91% to 97%. Literature suggests that contemporary SARS-CoV-2 VOC associated with immune escape affect viral neutralising capacity by 6–10 fold when compared to WT SARS-CoV-2[44–46] in vaccinated participants. Using this assumption, we considered a host NT50 of ≥1000 IU against WT SARS-CoV-2 would retain a robust neutralising capacity of at least 100 IU against commonly described VOC. For the linear regression model, we included only the later time point in individuals with repeated measures, to satisfy the assumptions of linear regression. Receiver operating curves, the area under the curve and the Youden Index were constructed using the pROC package in R[47]. We calculated sensitivity, specificity, positive and negative predictive values and overall accuracy, along with the binomial exact 95% confidence intervals of these thresholds to predict NT50 < 1000 IU against WT SARS-CoV-2, in different groups. We compared quantitative variables with the Kruskal–Wallis test and qualitative variables with the chi-square test. Univariate and multivariate linear regression models were constructed to determine the relationship between clinical variables and neutralising capacity. All statistical analysis was performed with R software, version 3.6.2.

### Reporting summary

Further information on research design is available in the Nature Portfolio Reporting Summary linked to this article.

## Data availability

Source data are provided in this paper. Additional participant data from the All Ireland Infectious Diseases Cohort study can be requested from the All-Ireland Infectious Diseases Cohort Study group. Data and samples are accessed through standardised data access guidelines and all approved data access requests are approved by a local ethics committee. Data from the EU-COVAT-1 study can be requested from the when the study is completed from the VACCELERATE - EU-COVAT-1 Part A Study Group. Viral isolate sequences are available at the following accession numbers: WT-B.1.177.18 (CEPHR_IE_B.177.18B.1.177.18_1220), GenBank accession ON350866; Beta (SARS-CoV-2/human/IRL/AIIDV1752/2021), GenBank: ON350868.1; Omicron-BA.5 (Pango lineage BA.5), GenBank accession OP508004. Source data are provided in this paper.

## Code availability

The analyses and graphics in this analysis were made using R version 3.6.2. Receiver operating characteristic analyses were constructed using the *pROC* package version 1.18.0.

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

## Acknowledgements

The authors wish to thank all study participants and their families for their participation and support in the conduct of the All Ireland Infectious Diseases Cohort Study and the VACCELERATE–EU-COVAT-1 Part A Study Group. The research leading to these results was conducted as part of the VACCELERATE consortium. For further information please refer to www.vaccelerate.eu. This investigation was conducted within the framework of VACCELERATE. VACCELERATE is funded by the European Union's Horizon 2020 research and innovation programme under grant number 101037867. This work was additionally supported by Science Foundation Ireland (grant number 20/COV/0305), the European Virus Archive GLO-BAL (EVA-GLOBAL) project that has received funding from the European Union's Horizon 2020 research and innovation programme under the grant agreement No 871029 and a philanthropic donation from Smurfit Kappa. G.K. was funded through a fellowship from the United States Embassy in Ireland during this study. S.O.R. is the recipient of the Irish Research Council (IRC) Government of Ireland Postgraduate Scholarship (GOIPG/2019/4432). Views and opinions expressed are those of the authors and do not necessarily reflect those of the European Union.

## Author contributions

G.K. conceived the study, recruited participants, performed the analyses and wrote the manuscript. S.O.R. conceived the study, performed the neutralising experiments and wrote the manuscript. N.W.K., D.O.'S. and A.H. contributed to the antigen-specific T-cell analysis. R.N., C.G., D.A., G.S. and A.A.G.L. processed the biobanked samples and performed the serologic assays. T.A., N.F. and M.A. assisted with the neutralising assays. W.T. supervised the statistical analyses. E.F., A.G., E.d.B., O.Y., M.H., P.D. and P.M. oversaw participant recruitment and clinical data acquisition. J.S., P.K. and O.C. lead the EU-COVAT-1-AGED Part A trial. R.C., O.O., J.D.L., O.C. and P.M. lead the VACCELERATE consortium. A.L. contributed to the design and analysis of the study. V.G. conceived the study and supervised the neutralising assays and the analyses. P.M. conceived the study and supervised the analyses. All authors reviewed and approved the final manuscript.

## Competing interests

E.F. has received consulting fees from Gilead, ViiV and Vidacare Ireland, and has been awarded a grant from Science Foundation Ireland outside the submitted work. E.d.B. has received consulting fees from Sanofi Pasteur and an honoraria/travel grant from Pfizer. P.M. has received honoraria and/or travel grants from Gilead Sciences, MSD, Astrazeneca, and ViiV Healthcare, and has been awarded grants by Science Foundation Ireland outside the submitted work. P.K. reports grants or contracts from German Federal Ministry of Research and Education (BMBF) B-FAST (Bundesweites Forschungsnetz Angewandte Surveillance und Testung) and NAPKON (Nationales Pandemie Kohorten Netz, German National Pandemic Cohort Network) of the Network University Medicine (NUM) and the State of North Rhine-Westphalia; Consulting fees Ambu GmbH, Gilead Sciences, Mundipharma Research Limited, Noxxon N.V. and Pfizer Pharma; Honoraria for lectures from Akademie für Infektionsmedizin e.V., Ambu GmbH, Astellas Pharma, BioRad Laboratories Inc., Datamed GmbH, European Confederation of Medical Mycology, Gilead Sciences, GPR Academy Ruesselsheim, HELIOS Kliniken GmbH, Lahn-Dill-Kliniken GmbH, medupdate GmbH, MedMedia GmbH, MSD Sharp & Dohme GmbH, Pfizer Pharma GmbH, Scilink Comunicación Científica SC, streamedup! GmbH and University Hospital and LMU Munich; Participation on an Advisory Board from Ambu GmbH, Gilead Sciences, Mundipharma Research Limited and Pfizer Pharma; A pending patent currently reviewed at the German Patent and Trade Mark Office (DE 10 2021 113 007.7); Other non-financial interests from Elsevier, Wiley

and Taylor & Francis online outside the submitted work. J.S. has received research support from the German Federal Ministry of Education and Research (BMBF) and Basilea Pharmaceuticals Inc.; has received speaker honoraria by Pfizer Inc., Gilead, and AbbVie; has been a consultant to Gilead, Produkt&Markt GmbH, Alvea Vax. and Micron Research and has received travel grants by German Society for Infectious Diseases (DGI e.V.) and Meta-Alexander Foundation. O.A.C. reports grants or contracts from BMBF, Cidara, EU-DG RTD (101037867), Pfizer; Consulting fees from Biocon, Biosys, Janssen, Noxxon, Pfizer; Honoraria for lectures from Gilead, MedScape, Merck/MSD, Pfizer; Participation on a Data Safety Monitoring Board or Advisory Board from Cidara, Janssen. A.L. receives consulting fees from Abbott and is co-chair of the Gilead Research Scholars Programme. All other authors declare no competing interests.

## Additional information

[1]Centre for Experimental Pathogen Host Research (CEPHR), University College Dublin, Belfield, Dublin 4, Ireland. [2]St Vincent's University Hospital, Elm Park, Dublin 4, Ireland. [3]Kathleen Lonsdale Institute for Human Health Research, Maynooth University, Maynooth, Co Kildare, Ireland. [4]Department of Infectious Diseases, Mater Misericordiae University Hospital, Eccles St, Dublin 7, Ireland. [5]Department of Infectious Diseases, Beaumont Hospital, Beaumont, Dublin 9, Ireland. [6]Department of International Health and Tropical Medicine, Royal College of Surgeons in Ireland, Dublin, Ireland. [7]Endocrinology Department, Wexford General Hospital, Carricklawn, Wexford, Ireland. [8]Department of Infectious Diseases, Cork University Hospital, Wilton, Co Cork, Ireland. [9]School of Medicine, University College Dublin, Belfield, Dublin 4, Ireland. [10]University of Cologne, Faculty of Medicine and University Hospital Cologne, Department of Internal Medicine and University of Cologne, Faculty of Medicine Institute of Translational Research, Cologne Excellence Cluster on Cellular Stress Responses in Aging-Associated Diseases (CECAD), Cologne, Germany. [11]German Centre for Infection Research (DZIF), Partner Site Bonn-Cologne Department Cologne, Cologne, Germany. [12]Influenza Centre, Department of Clinical Science, University of Bergen, Bergen, Norway. [13]European Vaccine Initiative, Heidelberg, Germany. [14]Department of internal Medicine, Rush University, Chicago, IL, USA. [15]National Children's Research Centre, Dublin 12, Ireland. [16]Vaccine Research Institute, Université Paris Est Créteil, Paris, France. [17]These authors contributed equally: Grace Kenny, Sophie O'Reilly. [18]A full list of members and their affiliations appears in the Supplementary Information. *Lists of authors and their affiliations appear at the end of the paper. ✉e-mail: grace.kenny1@ucd.ie

## The All Ireland Infectious Diseases Cohort Study

Grace Kenny ®[1,17]✉, Riya Negi[1], Colette Gaillard ®[1], Dana Alalwan[1], Gurvin Saini[1], Alejandro Garcia Leon[1], Eoin Feeney[1,2], Aoife G. Cotter[1,4], Eoghan de Barra[5,6], Obada Yousif[7], Mary Horgan[8] & Patrick W. G. Mallon[1,2]

## VACCELERATE Consortium EU-COVAT-1-AGED Part A Study Group

Jannik Stemler[10,11], Alejandro Garcia Leon[1], Patrick Mallon[1,2], Riya Negi[1], Colette Gaillard ®[1], Gurvin Saini[1], Philipp Koehler ®[10,11] & Oliver A. Cornely ®[10,11]

