## [Peer Review File · Nature Communications]

Reviewers' comments:

Reviewer #1 (Remarks to the Author):

Authors

Kenny et al study test concordance between in-lab nAb and binding ELISAs for SARS-CoV-2. It is thought that Ab is functionally important for vaccine-induced protection and Ab assays were being used to benchmark vaccines and to some extent to guide patient care for extra doses of vaccine. Strengths of the paper include samples from a variety of times after infection, inclusion of clinical metadata, inclusion of a WHO standard, and crisp writing. Limitations include use of samples from only early in the pandemic, use of test antigens and viruses for nAb that do not reflect current VOC, exclusion of vaccine or hybrid immunity specimens, and the use of in-house ELISAs rather than commercially available tests. In addition, a good correlation between binding IgG to various regions of S, and nAb, is already well established in the literature, and it is not surprising biologically that binding Ab levels to RBD would correlate best with nAb. There is also not a "cutoff" value for nAb considered to be a correlate of protection. Overall, the levels of conceptual or technical novelty or of medical Significance seem modest and the report might be better for a diagnostics specialty venue.

For infections such as HBV and influenza, there are antibody "levels" that are accepted as "protective". The authors remain silent on whether there is a similar metric for SARS CoV 2. We generally think higher is better and broader (also active against VOC) is better. The authors discuss on page 3 that there is a lack of a validated threshold of immunity. They start using a threshold of 1:1000 on page 5, without saying why, and extend this to 1:100 for beta, again without justifying the medical meaning of this level. Mention is made in Discussion of a RBD binding level of 506 IU/ml as a potential CoP after vaccination, so it is curious they did not evaluate the ROC AUC for their assay at this level. Without such as Correlate of Protection (CoP), the Significance of a person having a specified level of nAb is uncertain. There is a rich literature on candidate CoP for SARS CoV 2 and it is suggested that the authors place their findings in this context.

The T cell parameter checked, IFN-g, is not relevant to CD4 T cell help to antibody responses. The levels of so-called circulating T follicular helper cells (cTFH) can be estimated by candidate flow cytometry-based assays.

The authors rightly state that true nAb assays are expensive, slow, and need BSL3. There are myriad papers that use BSL2 level pseudovirus neutralization (PsV nAb) with rapid readouts such as fluorescence, luminescence, etc., and there are commercial assays that measure the inhibition of S or RBD binding to ACE2, using defined recombinant proteins, as a surrogate for nAb. These tests are variably cheaper, faster and safer than true nAb assays. The paper would be stronger if the authors compared the sensitivity/specificity/ROC AUC of the their candidate to these widely used tests and discussed why and how their candidate is better than these alternatives.

The statements about anti-S antigen levels declining and then plateauing, in contrast to anti-N continuing to decline (P4, 5) are not supported statistically. Can the authors clarify if they are claiming there is a statistically significant difference in the kinetics between these assays or if they are just being descriptive about the trend lines shown in Fig 1ABC vs. Fig 1D? If they do assert significance, please state statistical methods and p values.

No data from seronegative persons are provided for the T cell assay as negative controls.

Minor:

What is the definition of lower limit of quantification in Fig. 1?

Nature family journals tend to require examples of flow cytometry gating trees to allow evaluation of flow-based data. These are not provided for CD69 or PD1 so it is hard to evaluate these data.

Were the technical replicates mentioned for the flow nAb assay duplicates, triplicates, etc.?

A little more info is needed on the T cell assay. What is the final concentration of each peptide, how long are they? What chemical identity is used to inhibit protein export?

Asterisk in Supp Tab 2 needs to be explained

Reviewer #2 (Remarks to the Author):

In this manuscript, Kenny, O'Reilly, and colleagues determine plasma SARS-CoV-2 binding and neutralizing activity as well as T cell reactivity in a cohort of 131 convalescent individuals. By correlating binding and neutralizing titers, they propose a cut-off for RBD-binding activity (that can be readily determined) that indicates the presence of plasma neutralizing activity in lieu of more laborious assessments of neutralizing activity. The manuscript is well written and approachable.

While the correlation between anti-SARS-CoV-2 binding titers and neutralizing activity has been demonstrated numerous times, this study adds an analysis (performed less frequently in other studies) that determines a specific anti-RBD titer as indicative of a particular neutralizing titer. Because this neutralizing titer has been determined against the very antibody-sensitive wild-type variant, the relevance of the anti-RBD cut-off identified in this study is likely, however, to be limited in the era of the much more resistant Omicron variants.

Major points:

a.) The title should be limited to prediction of “robust” host neutralizing activity against the wild-type variant.

b.) In their description of neutralization results and the abstract, it should be made clearer that the NT50 titers used for the majority of the analysis were obtained against the wild-type variant (which the convalescent individuals were likely infected with).

c.) Discussion/Paragraph 3: While some work by others on correlation analyses of binding and neutralizing titers is cited, reference to and discussion to more similar work (including work determining binding titers predictive of neutralization) appears to be missing. For example: 10.1515/cclm-2021-0700; 10.3389/fcimb.2022.822599; 10.1002/jmv.27287

d.) Section “Comparison of T cell responses to RBD titer”: Data should be shown - otherwise this analysis is of limited value.

e.) The methods section suggests that three different viruses were used (a very early wild-type variant without D614G (2019-nCoV/Italy-INMI1), a D614G wild-type variant, and the Beta variant). However, the text only mentions the D614G variant when discussing the wild-type virus - suggesting that the variant without D614G was not at all used. Because D614G affects susceptibility to neutralizing antibodies, this should be clarified. Similarly, two different cell types are described as being used in the methods (VeroE6 and VeroE6/TMPRSS2), which is not clear from the main text. Again, the choice of the cell line can affect the outcome of neutralization assays. To ensure that the assay used was consistent and results can be adequately compared, it should be made clear which cell line and which (wild-type) virus was used in which situation.

f.) The rationale for the selection of an NT50 titer of 1:1000 as “target” should be made clear in the main text.

More minor comments:

g.) Section “Relationship between NT50 and IgG”/ Paragraph 1: The correlation between NT50 and antigen binding beyond RBD and S1 should be stated and could also be shown in graphical form (perhaps in supplement; also applies to correlation after restriction individuals >30 d post symptom-onset).

h.) Section “Relationship between NT50 and IgG”/ Paragraph 2: While sensitivity and specificity of RBD for NT50 prediction are mentioned in the text, what are the results for the other antigens?

i.) Section “Relationship between NT50 and IgG”: Only the correlation of RBD and NT50 is shown in a figure. The correlation for the other antigens as well as the correlation after restriction to individuals >30 d post-symptom onset could also be shown in graphics (perhaps in supplement).

j.) Discussion/Paragraph 2: The authors state that individuals with an RBD titer >456 IU/ml have measurable SARS-CoV-2-specific T cell responses. While this is technically correct, their results section on T cell immunity indicates that there is no detectable difference in T cell immunity between individuals with anti-RBD titers below or above this cut-off (suggesting this cut-off is of no value to assess T cell immunity). This should be made clear to avoid confusion.

k.) Discussion/Paragraph 4: When stating that “other assays demonstrated weaker correlations”, this should be backed by references.

l.) Discussion/Paragraph 5: Beyond potential differences in expression levels of the spike protein on the viral surface between pseudoviruses and live viruses, differences in read-out methods and target cells can explain differences seen between different assay types. This could be added.

m.) 190 samples from 131 individuals were analyzed. It would be helpful to add how many individuals provided how many samples (xx individuals contributed 1 samples, xx individuals contributed 2 samples, xx individuals contributed 3 samples, etc.).

n.) Discussion/Paragraph 9: The reference in the sentence “...estimated to provide 80% vaccine efficacy against symptomatic SARS-CoV-2 (typo) on study” should likely be changed from 26 to 27. The live virus neutralization titer associated with 80% VE and the similar anti-RBD titer is quite distinct from the NT50 described here (264 vs. >1:1,000). Can the authors comment on this?

o.) It would be helpful for the review process to add line numbers.

Reviewer #3 (Remarks to the Author):

The study from Kenny, Reilly and their colleagues defined a threshold of RBD binding titers to predict the neutralizing capacity, which is potentially helpful to be used as a protective indicator for clinical evaluation. This study might be of some interest, however, there are some concerns.

1. As the author indicated in the discussion, this study mainly focused on those who were naturally infected and recovered, as shown by many studies, the dynamics of the antibody titers (against RBD, NTD or S2) varied in individuals that were exposed with different antigens, e.g. different type of vaccines, original strain and the emerging variants. Besides, the authors tried to use the threshold of 456IU/ml of RBD titers as a predictor for both infection with original strain or Beta variant, this is not quite convincing since we know that both binding titers and neutralizing activity (plasma or mAbs) against Beta variant reduced significantly among convalescents or vaccinees. In this case, the author may need to use a different threshold of binding titers against Beta RBD, and check how it correlates with infection or diseases.
2. As now the main circulating variants changed to Omicron and its subvariants, which are the main cause for the current pandemic, the author should check if the threshold of RBD titer hold true for the neutralizing capacity against Omicron, which seems very unlikely. However, the author may can explore the Omicron BA. 4/5 RBD titer threshold.
3. As evidenced in many studies, in the early stage of the infection or the onset of disease, not only IgG but IgM or IgA were well correlated with the plasma neutralizing activity, it might be good to check those isotypes as well.
4. Memory B and T cells are important components for rapid responding to the next pathogen encountering. Since the donors have been discharged for 1-6 months after infection, it's better to check the memory B and T cell populations in the PBMC, to see if their responses can be correlated with the RBD titer threshold.
5. For the detection of T cells responses, the authors should stain CD8a or CD8b to distinguish the CD8 T cell population. Also, it would be interesting to check the Tfh cell population because it is known to provide B-cell help for the induction of antigen-specific antibody production in COVID.
6. Can the authors provide explanations for the association between history of cardiac disease/hypertension and NT50>1000?
7. There are some typos in the line138-line152, e.g. line143, p 0.17.

Reviewer #1 (Remarks to the Author):

Authors

Kenny et al study test concordance between in-lab nAb and binding ELISAs for SARS-CoV-2. It is thought that Ab is functionally important for vaccine-induced protection and Ab assays were being used to benchmark vaccines and to some extent to guide patient care for extra doses of vaccine. Strengths of the paper include samples from a variety of times after infection, inclusion of clinical metadata, inclusion of a WHO standard, and crisp writing. Limitations include use of samples from only early in the pandemic, use of test antigens and viruses for nAb that do not reflect current VOC, exclusion of vaccine or hybrid immunity specimens, and the use of in-house ELISAs rather than commercially available tests.

In this new manuscript, we have significantly expanded the analysis to include two additional cohorts. One cohort who had received a primary two dose vaccine series, and an additional group more reflective of current immunity with a history of both omicron period infection and at least one booster vaccine dose. We present additional experiments, including neutralising capacity against the Omicron BA.5 variant. While we again use an in house ELISA, this has been validated against two different commercial assays (doi: 10.1016/j.jim.2022.113345).

In addition, a good correlation between binding IgG to various regions of S, and nAb, is already well established in the literature, and it is not surprising biologically that binding Ab levels to RBD would correlate best with nAb. There is also not a “cutoff” value for nAb considered to be a correlate of protection. Overall, the levels of conceptual or technical novelty or of medical Significance seem modest and the report might be better for a diagnostics specialty venue. For infections such as HBV and influenza, there are antibody “levels” that are accepted as “protective”. The authors remain silent on whether there is a similar metric for SARS CoV 2. We generally think higher is better and broader (also active against VOC) is better. The authors discuss on page 3 that there is a lack of a validated threshold of immunity. They start using a threshold of 1:1000 on page 5, without saying why, and extend this to 1:100 for beta, again without justifying the medical meaning of this level. Mention is made in Discussion of a RBD binding level of 506 IU/ml as a potential CoP after vaccination, so it is curious they did not evaluate the ROC AUC for their assay at this level. Without such as Correlate of Protection (CoP), the Significance of a person having a specified level of nAb is uncertain. There is a rich literature on candidate CoP for SARS CoV 2 and it is suggested that the authors place their findings in this context.

In order to place this analysis within the broader literature, we normalised the neutralising assay to the WHO standard (in the previous manuscript only the ELISA was normalised to the WHO international standard). We were then able to use thresholds which have been shown to be associated with protection against infection in two independent vaccine trials (DOI: 10.1126/science.abm3425, <https://doi.org/10.1038/s41467-022-35768-3>). The nAb thresholds used in this new analysis are 1000 IU and 100 IU, which due to the conversion factor are higher

thresholds than the 1:1000 and 1:100 used in the previous manuscript. We reanalysed the data in the previous manuscript using these new thresholds, and also applied them to the two additional cohorts and examined BA.5 in addition to the Beta variant.

The T cell parameter checked, IFN-g, is not relevant to CD4 T cell help to antibody responses. The levels of so-called circulating T follicular helper cells (cTFH) can be estimated by candidate flow cytometry-based assays.

While we agree that cTFH are important for antibody responses following infection and vaccination, intracellular cytokine staining is often considered the gold standard for determining functional antigen specific T cell response. The primary goal of the T cell analysis was to determine the correlation between cellular immunity and nAb response, rather than T cell help in producing antibody response.

The authors rightly state that true nAb assays are expensive, slow, and need BSL3. There are myriad papers that use BSL2 level pseudovirus neutralization (PsV nAb) with rapid readouts such as fluorescence, luminescence, etc., and there are commercial assays that measure the inhibition of S or RBD binding to ACE2, using defined recombinant proteins, as a surrogate for nAb. These tests are variably cheaper, faster and safer than true nAb assays. The paper would be stronger if the authors compared the sensitivity/specificity/ROC AUC of the their candidate to these widely used tests and discussed why and how their candidate is better than these alternatives.

While it is true that the stated assays are quicker and easier than a live virus neutralising assay, this paper is making the case for the use of a binding antibody assay, rather than for the widespread use of the neutralising assay.

The statements about anti-S antigen levels declining and then plateauing, in contrast to anti-N continuing to decline (P4, 5) are not supported statistically. Can the authors clarify if they are claiming there is a statistically significant difference in the kinetics between these assays or if they are just being descriptive about the trend lines show in Fig 1ABC vs. Fig 1D? If they do assert significance, please state statistical methods and p values.

We have removed this section of the paper as we believe it does not contribute from the main message of the paper, and there the analyses of the new cohorts are presented in its place.

No data from seronegative persons are provided for the T cell assay as negative controls.

Unstimulated cells serve as the negative controls in this assay, as previously described (<https://doi.org/10.1002/oby.23526>).

Minor:

What is the definition of lower limit or quantification in Fig. 1?

This figure has been removed in the new manuscript.

Nature family journals tend to require examples of flow cytometry gating trees to allow evaluation of flow-based data. These are not provided for CD69 or PD1 so it is hard to evaluate these data.

These have now been included.

Were the technical replicates mentioned for the flow nAb assay duplicates, triplicates, etc.?

This section of the methods has been abbreviated as there is now reference to a detailed paper describing development and validation of the neutralising assay.

A little more info is needed on the T cell assay. What is the final concentration of each peptide, how long are they? What chemical identity is used to inhibit protein export?

These have been added.

Asterisk in Supp Tab 2 needs to be explained

This table has been removed.

Reviewer #2 (Remarks to the Author):

In this manuscript, Kenny, O'Reilly, and colleagues determine plasma SARS-CoV-2 binding and neutralizing activity as well as T cell reactivity in a cohort of 131 convalescent individuals. By correlating binding and neutralizing titers, they propose a cut-off for RBD-binding activity (that can be readily determined) that indicates the presence of plasma neutralizing activity in lieu of more laborious assessments of neutralizing activity. The manuscript is well written and approachable.

While the correlation between anti-SARS-CoV-2 binding titers and neutralizing activity has been demonstrated numerous times, this study adds an analysis (performed less frequently in other studies) that determines a specific anti-RBD titer as indicative of a particular neutralizing titer. Because this neutralizing titer has been determined against the very antibody-sensitive wild-type variant, the relevance of the anti-RBD cut-off identified in this study is likely, however, to be limited in the era of the much more resistant Omicron variants.

Major points:

a.) The title should be limited to prediction of "robust" host neutralizing activity against the wild-type variant.

This analysis now includes two additional cohorts of individuals, a cohort that received a primary two dose vaccine series, and a an additional group more

reflective of current immunity with a history of both omicron period infection and at least one booster vaccine dose. These cohorts had neutralizing activity assessed against the Beta variant and Omicron BA.5 variant respectively. We have also normalised the neutralising assay to allow us to use neutralising thresholds associated with protection against infection in vaccine trials, and repeated the analysis with these new thresholds. The title has been changed to better reflect this new analysis.

b.) In their description of neutralization results and the abstract, it should be made clearer that the NT50 titers used for the majority of the analysis were obtained against the wild-type variant (which the convalescent individuals were likely infected with).

We have performed additional analyses with both Beta and BA.5 variants to address this concern.

c.) Discussion/Paragraph 3: While some work by others on correlation analyses of binding and neutralizing titers is cited, reference to and discussion to more similar work (including work determining binding titers predictive of neutralization) appears to be missing. For example: 10.1515/cclm-2021-0700; 10.3389/fcimb.2022.822599; 10.1002/jmv.27287

These references have been added and discussed.

d.) Section “Comparison of T cell responses to RBD titer”: Data should be shown - otherwise this analysis is of limited value.

As we found no significant findings within the T cell analysis, we omitted these data as we did not believe these contributed to the overall message of the manuscript.

e.) The methods section suggests that three different viruses were used (a very early wild-type variant without D614G (2019-nCoV/Italy-INMI1), a D614G wild-type variant, and the Beta variant). However, the text only mentions the D614G variant when discussing the wild-type virus - suggesting that the variant without D614G was not at all used. Because D614G affects susceptibility to neutralizing antibodies, this should be clarified. Similarly, two different cell types are described as being used in the methods (VeroE6 and VeroE6/TMPRSS2), which is not clear from the main text. Again, the choice of the cell line can affect the outcome of neutralization assays. To ensure that the assay used was consistent and results can be adequately compared, it should be made clear which cell line and which (wild-type) virus was used in which situation.

This has been clarified in this new analysis – WT without D614G is referred to as WT-B throughout the text and D614G WT variant referred to as WT-B.1.177.18 throughout the text.

f.) The rationale for the selection of an NT50 titer of 1:1000 as “target” should be made clear in the main text.

We have reanalysed the previous data and all analysed all new additional data with two clinically relevant thresholds derived from two independent vaccine trials (DOI: 10.1126/science.abm3425, <https://doi.org/10.1038/s41467-022-35768-3>). We normalised the neutralising assay to the WHO standard (in the previous manuscript only the ELISA was normalised to the WHO international standard) in order to do this. The neutralising thresholds used in this new analysis are 1000 IU and 100 IU, which due to the conversion factor are higher thresholds than the 1:1000 and 1:100 used in the previous manuscript.

More minor comments:

g.) Section “Relationship between NT50 and IgG”/ Paragraph 1: The correlation between NT50 and antigen binding beyond RBD and S1 should be stated and could also be shown in graphical form (perhaps in supplement; also applies to correlation after restriction individuals >30 d post symptom-onset).

This has been added in the supplement.

h.) Section “Relationship between NT50 and IgG”/ Paragraph 2: While sensitivity and specificity of RBD for NT50 prediction are mentioned in the text, what are the results for the other antigens?

This has been added in the supplement.

i.) Section “Relationship between NT50 and IgG”: Only the correlation of RBD and NT50 is shown in a figure. The correlation for the other antigens as well as the correlation after restriction to individuals >30 d post-symptom onset could also be shown in graphics (perhaps in supplement).

Spearman correlation for all antigens has been added in the supplement.

j.) Discussion/Paragraph 2: The authors state that individuals with an RBD titer >456 IU/ml have measurable SARS-CoV-2-specific T cell responses. While this is technically correct, their results section on T cell immunity indicates that there is no detectable difference in T cell immunity between individuals with anti-RBD titers below or above this cut-off (suggesting this cut-off is of no value to assess T cell immunity). This should be made clear to avoid confusion.

k.) Discussion/Paragraph 4: When stating that “other assays demonstrated weaker correlations”, this should be backed by references.

l.) Discussion/Paragraph 5: Beyond potential differences in expression levels of the spike protein on the viral surface between pseudoviruses and live viruses, differences in read-out methods and target cells can explain differences seen between different assay types. This could be added.

For points j-l – discussion has been changed significantly to reflect the substantial additional analysis and all of these points have been removed.

m.) 190 samples from 131 individuals were analyzed. It would be helpful to add how many individuals provided how many samples (xx individuals contributed 1 samples, xx individuals contributed 2 samples, xx individuals contributed 3 samples, etc.).

This has been added in the legend of the table 1.

n.) Discussion/Paragraph 9: The reference in the sentence “...estimated to provide 80% vaccine efficacy against symptomatic SARS-CoV-2 (typo) on study” should likely be changed from 26 to 27. The live virus neutralization titer associated with 80% VE and the similar anti-RBD titer is quite distinct from the NT50 described here (264 vs. >1:1,000). Can the authors comment on this?

o.) It would be helpful for the review process to add line numbers.

These have been included.

Reviewer #3 (Remarks to the Author):

The study from Kenny, Reilly and their colleagues defined a threshold of RBD binding titers to predict the neutralizing capacity, which is potentially helpful to be used as a protective indicator for clinical evaluation. This study might be of some interest, however, there are some concerns.

- 1. As the author indicated in the discussion, this study mainly focused on those who were naturally infected and recovered, as shown by many studies, the dynamics of the antibody titers (against RBD, NTD or S2) varied in individuals that were exposed with different antigens, e.g. different type of vaccines, original strain and the emerging variants. Besides, the authors tried to use the threshold of 456IU/ml of RBD titers as a predictor for both infection with original strain or Beta variant, this is not quite convincing since we know that both binding titers and neutralizing activity (plasma or mAbs) against Beta variant reduced significantly among convalescents or vaccinees. In this case, the author may need to use a different threshold of binding titers against Beta RBD, and check how it correlates with infection or diseases.*
- 2. As now the main circulating variants changed to Omicron and its subvariants, which are the main cause for the current pandemic, the author should check if the threshold of RBD titer hold true for the neutralizing capacity against Omicron, which seems very unlikely. However, the author may can explore the Omicron BA. 4/5 RBD titer threshold.*

We have substantially expanded the analysis to include 2 additional cohorts of individuals, a cohort that received a primary two dose vaccine series, and a cohort with a history of infection in the Omicron dominant period that had received a booster vaccine. We evaluated neutralising capacity against Beta and Omicron in these two cohorts respectively, and explored the performance RBD thresholds within these new cohorts.

3. As evidenced in many studies, in the early stage of the infection or the onset of disease, not only IgG but IgM or IgA were well correlated with the plasma neutralizing activity, it might be good to check those isotypes as well.

We agree that IgM and IgA responses are important components of the overall immune response to SARS-CoV-2, we aimed to find a single, easily measurable correlate of protection, and as IgG is more stable than IgM or IgA (Qi et al 2022 <https://doi.org/10.1038/s41590-022-01248-5>), focused only on this isotype.

4. Memory B and T cells are important components for rapid responding to the next pathogen encountering. Since the donors have been discharged for 1-6 months after infection, it's better to check the memory B and T cell populations in the PBMC, to see if their responses can be correlated with the RBD titer threshold.

5. For the detection of T cells responses, the authors should stain CD8a or CD8b to distinguish the CD8 T cell population. Also, it would be interesting to check the Tfh cell population because it is known to provide B-cell help for the induction of antigen-specific antibody production in COVID.

Reply to point 4 and 5: We agree that memory B and T cells are important components of immune memory. While we did examine functional T cell response in this manuscript, further T and B cell work was beyond the scope of this analysis.

6. Can the authors provide explanations for the association between history of cardiac disease/hypertension and NT50>1000?

This section of the prior manuscript has been removed.

7. There are some typos in the line138-line152, e.g. line143, p 0.17.

The manuscript has been revised so these have been addressed.

REVIEWER COMMENTS

Reviewer #1 (Remarks to the Author):

Kenny et al. have substantially revised the manuscript, most notably by adding new cohorts, adding data for omicron for one of the new cohorts (hybrid immunity), and deleting CMI data. As before it is crisply written. Technical details on assays have been provided as requested.

Reviewer #2 (Remarks to the Author):

In this revised version of their manuscript, Kenny, O'Reilly and colleagues aim to identify an anti-RBD-IgG serum level as a correlate for a SARS-CoV-2 neutralizing serum titer that may be associated with protection from symptomatic infection. Compared to the original version of the manuscript, the authors have added analyses of samples from diverse additional patient cohorts (vaccinated and hybrid immunity). For a subset of samples, they now also include results of neutralization assays against the Beta and BA.5 variants. Using the RBD titer determined in unvaccinated convalescent individuals following neutralization assays against wild type virus, the authors go on to show that the RBD cut-off can be useful in excluding low neutralizing activity against the Beta and/or BA.5 variants in different cohorts.

Including analyses of the RBD titer against more recently emerged SARS-CoV-2 variants associated with immune escape strengthens the manuscript. As the authors acknowledge, the applicability of distinct WT RBD titers to be predictive of protection in the future may be limited (e.g., due to increasing use of bivalent and/or non WT-based vaccines).

Main comments:

a)The authors now cite three immune correlate analyses of vaccine trials (Chadox1, Feng et al.; Moderna, Gilbert et al.; Novavax, Fong et al.) that have previously correlated neutralizing activity with vaccine efficacy. A similar study on the Ad26.COVS vaccine (Fong et al., Nat Microbiol 2022) should also be included. All of these studies also provide a correlation for RBD antibody levels with vaccine efficacy, as is indirectly attempted in the current manuscript. The results from these earlier works on RBD titers should be put more into perspective, including in the abstract in which the authors currently state that "what threshold ... predicts sufficient neutralizing activity ... remains unclear."

b) In the cited immune correlate analyses, the neutralization titers correlated with protection were determined using pseudovirus-based assays. In the current manuscript, a live virus-based microneutralization assay is used. How do these assays compare? This is important because the anti-RBD levels considered as potentially protective in the current manuscript are based on neutralization titers determined by the live-virus assay assuming they are reflective of vaccine efficacy determined for pseudovirus neutralization titers. The manuscript describing the neutralization assay used includes a comparison to a live virus plaque reduction test, but not to pseudovirus assays.

c) A whole paragraph of the results sections that refers to results of T cell analyses remains in the manuscript and the supplementary information also include an exemplary gating strategy for flow cytometry-based T cell assays. However, not a single data point is shown and all the authors indicate is that there were “no significant differences” and “no correlation” with the addition “(p >0.05, data not shown)”. As for the first version of the manuscript, my opinion remains that either the data should be shown or the paragraph should be removed.

d) The capacity of the RBD titer to predict neutralization against immune escape variants is only tested against the Beta variant for the two cohorts of unvaccinated convalescent individuals and individuals after the completion of the primary immune series. How would testing the RBD cut-off in predicting BA.5 neutralization look like (as was done only for the individuals with hybrid immunity)? My assumption is that its use would be quite limited here as there may be almost no detectable BA.5 neutralizing activity in these settings despite robust WT RBD titers. This comment also pertains to the first paragraph of the discussion, in which the authors state that RBD threshold retains accuracy when tested in diverse populations.

e) The emergence of novel variants with even higher levels of immune escape may compromise the applicability of the (wild type-based) RBD threshold determined in this study and this should be discussed.

Minor comments:

f) The title appears incomplete. “Correlation of RBD and Neutralising Capacity” / “RBD threshold” – a word after RBD (e.g., “binding”) seems to be lacking.

g) Is there any data showing that the serum neutralization titers considered as protective in the early immune correlate analyses (neutralization against wild-type in the setting of wild-type exposure) hold true for the newer variants? In other words, would i.) a neutralization titer of 100 against BA.5 show the

same level of protection against BA.5 infection as ii.) a WT neutralization titer of 100 does against WT infection?

h) Lines 92-94: What do the authors consider as 'excellent' vaccine efficacy (VE)? Immune correlate analyses providing distinct IU50 nAb levels for different levels of VE were also performed for the Chadox-1 vaccine (Feng et al, Nat Med 2021) and the Ad26.COV2.S platform (Fong et al., Nat Microbiol 2022) and these should also be included in this section of the paragraph.

i) The authors repeatedly mention that the correlation between NT50 and RBD titers is weaker in the early post-infection period than in the later convalescent phase (e.g., lines 145-147 and lines 183-185). This is not entirely unexpected because (for example) IgM antibodies will contribute to neutralization of serum early after infection (and less so late after infection) but will not be detected in the RBD assay that only identifies IgG antibodies.

Reviewer #3 (Remarks to the Author):

Thank you for the rebuttal. I have to say, the new version is improved.

However, I still have some concerns, which may not invalidate the findings of the study but suggest areas for further research and consideration.

1. The study primarily focuses on individuals who experienced acute symptoms of COVID-19 in the early stages of the pandemic or those who received vaccinations during a specific time period. However, there is a lack of long-term data on the dynamics of neutralizing antibodies and their correlation with RBD (receptor binding domain) titers.

2. The study acknowledges the limitations of neutralizing assays, particularly live virus assays, due to their labor-intensive nature, high cost, and the requirement for high biosafety level facilities. These limitations hinder their use in large-scale vaccine trials or routine clinical settings. Additionally, there is significant inter-individual variation in the correlation between neutralizing antibodies and anti-spike IgG binding antibodies, which may impact the accuracy of predicting protective immunity solely based on binding antibody assays.

3. Furthermore, the study recognizes that the immune response to SARS-CoV-2 can vary depending on several factors, such as the individual's infection history, vaccine platform, number of vaccine doses, and time intervals between doses or previous infections. These factors contribute to the heterogeneity of population immunity, making it challenging to establish a universal threshold for protective immunity based solely on RBD titers.

4. The authors proposed a discrete RBD threshold as a predictor of protective neutralizing capacity against different variants of concern (VOC). However, it is essential to externally validate these findings in independent cohorts to ensure the reliability and generalizability of the proposed threshold.

REVIEWER COMMENTS

Reviewer #1 (Remarks to the Author):

Kenny et al. have substantially revised the manuscript, most notably by adding new cohorts, adding data for omicron for one of the new cohorts (hybrid immunity), and deleting CMI data. As before it is crisply written. Technical details on assays have been provided as requested.

We thank the reviewer for these supportive comments.

Reviewer #2 (Remarks to the Author):

In this revised version of their manuscript, Kenny, O'Reilly and colleagues aim to identify an anti-RBD-IgG serum level as a correlate for a SARS-CoV-2 neutralizing serum titer that may be associated with protection from symptomatic infection. Compared to the original version of the manuscript, the authors have added analyses of samples from diverse additional patient cohorts (vaccinated and hybrid immunity). For a subset of samples, they now also include results of neutralization assays against the Beta and BA.5 variants. Using the RBD titer determined in unvaccinated convalescent individuals following neutralization assays against wild type virus, the authors go on to show that the RBD cut-off can be useful in excluding low neutralizing activity against the Beta and/or BA.5 variants in different cohorts.

Including analyses of the RBD titer against more recently emerged SARS-CoV-2 variants associated with immune escape strengthens the manuscript. As the authors acknowledge, the applicability of distinct WT RBD titers to be predictive of protection in the future may be limited (e.g., due to increasing use of bivalent and/or non WT-based vaccines).

Main comments:

a) The authors now cite three immune correlate analyses of vaccine trials (Chadox1, Feng et al.; Moderna, Gilbert et al.; Novavax, Fong et al.) that have previously correlated neutralizing activity with vaccine efficacy. A similar study on the Ad26.COVS vaccine (Fong et al., Nat Microbiol 2022) should also be included. All of these studies also provide a correlation for RBD antibody levels with vaccine efficacy, as is indirectly attempted in the current

manuscript. The results from these earlier works on RBD titers should be put more into perspective, including in the abstract in which the authors currently state that “what threshold ... predicts sufficient neutralizing activity ... remains unclear.”

We thank the reviewer for highlighting this relevant additional study and have added this reference in the introduction. We have also discussed the binding antibody analyses used in these prior studies in more detail in the introduction (page 3 line 13), and edited the abstract to put the current work in the context of these prior studies.

While we agree with the reviewer that the immune correlates studies of vaccine trials provide provisional evidence for binding IgG thresholds and protection against infection, the analyses presented in this study expand on these findings and provide a rationale for the use of a defined threshold in the era of immune evasive VOCs and a changing profile of host immunity, specifically hybrid immunity, as most vaccine trials were restricted to individuals who were naive to SARS-CoV-2 infection receiving primary series vaccination and evaluated immunity over a relatively short time point post vaccination. As such we consider the finding of the current studies to be novel and of current clinical relevance.

b) In the cited immune correlate analyses, the neutralization titers correlated with protection were determined using pseudovirus-based assays. In the current manuscript, a live virus-based microneutralization assay is used. How do these assays compare? This is important because the anti-RBD levels considered as potentially protective in the current manuscript are based on neutralization titers determined by the live-virus assay assuming they are reflective of vaccine efficacy determined for pseudovirus neutralization titers. The manuscript describing the neutralization assay used includes a comparison to a live virus plaque reduction test, but not to pseudovirus assays.

We thank the reviewer for highlighting this important issue. The current study utilises live-virus assays to determine viral neutralisation potential, and which are considered the gold standard for measuring viral neutralisation. However, they are not practical for large scale studies, or accessible to all research centres, resulting in a wide range of both live and pseudovirus assays being used across publications. While the correlations between pseudovirus and live virus assays have been shown to be high, the specific NT50 values produced can differ widely across assays. A major revision we have made to this manuscript from our initial submission is to standardise the readings from our live virus neutralisation assay to International Units, using the First WHO International Reference Panel for anti-

SARS-CoV-2 Immunoglobulin, NIBSC code: 20/268, designed to enable comparison of results from diverse assays. Many prior assays (including pseudovirus assays) have not included this standardisation. We have included these details in the method section. This standardisation allows us to directly compare the results of this study to the immune correlates analyses that normalised their assays to the same standard. However, we acknowledge that assay choice adds an additional variable and have highlighted this in the limitations (page 14 lines 19-21).

c) A whole paragraph of the results sections that refers to results of T cell analyses remains in the manuscript and the supplementary information also include an exemplary gating strategy for flow cytometry-based T cell assays. However, not a single data point is shown and all the authors indicate is that there were “no significant differences” and “no correlation” with the addition “(p >0.05, data not shown)”. As for the first version of the manuscript, my opinion remains that either the data should be shown or the paragraph should be removed.

We thank the reviewer for this feedback and have now added additional data on T cell outputs into the supplementary information.

d) The capacity of the RBD titer to predict neutralization against immune escape variants is only tested against the Beta variant for the two cohorts of unvaccinated convalescent individuals and individuals after the completion of the primary immune series. How would testing the RBD cut-off in predicting BA.5 neutralization look like (as was done only for the individuals with hybrid immunity)? My assumption is that its use would be quite limited here as there may be almost no detectable BA.5 neutralizing activity in these settings despite robust WT RBD titers. This comment also pertains to the first paragraph of the discussion, in which the authors state that RBD threshold retains accuracy when tested in diverse populations.

We thank the reviewer for this comment. Over the course of the pandemic, both population immunity and circulating variants have changed with time. The analyses presented within this manuscript aimed to test the relationship between RBD and viral neutralisation capacity across the evolution of the COVID19 pandemic, specifically by testing viral strains against plasma from subjects derived from the same period in which the specific viral strain was circulating.

The Beta variant was among the most evasive variants circulating at the time COVID19 vaccination was initially rolled out and so was selected as the variant to test in the convalescent and two dose vaccine groups. As circulation of the Beta variant was relatively low in Ireland, this group is assumed to have been naive to the Beta variant but the relationship between RBD and viral neutralising capacity against Beta variant persisted.

Although we acknowledge that testing neutralising capacity against BA.5 in an unvaccinated population or individuals who had received primary vaccination when the Beta variant was prevalent would have provided additional information to this study, our view was that the best population against which to test the BA.5 variant was the population prevalent during circulation of BA.5, which were predominantly vaccinated with or without hybrid immunity, in which we showed the relationship between RBD and neutralising capacity to be robust. However we acknowledge that BA.5 may have elicited very low levels of neutralising capacity when tested against plasma taken from individuals well before the emergence of this strain and have discussed these potential uncertainties within the limitations section of the discussion (page 14 line 6).

e) The emergence of novel variants with even higher levels of immune escape may compromise the applicability of the (wild type-based) RBD threshold determined in this study and this should be discussed.

We thank the reviewer for this comment and have added this discussion point on page 14 line 23.

Minor comments:

f) The title appears incomplete. "Correlation of RBD and Neutralising Capacity" / "RBD threshold" – a word after RBD (e.g., "binding") seems to be lacking.

We have added IgG to the title to amend this.

g) Is there any data showing that the serum neutralization titers considered as protective in the early immune correlate analyses (neutralization against wild-type in the setting of wild-type exposure) hold true for the newer variants? In other words, would i.) a neutralization

titer of 100 against BA.5 show the same level of protection against BA.5 infection as ii.) a WT neutralization titer of 100 does against WT infection?

We thank the reviewer for raising this interesting point. We are not aware of any studies that evaluate the protection against BA.5 associated with an NT50 of 100 IU against BA.5. However, a model that predicted vaccine efficacy against WT SARS-CoV-2 using neutralising data from the early vaccine trials, showed good agreement with efficacy studies against Alpha, Delta, Beta (Cromer et al. Lancet Microbe 2021) and Omicron (BA.1) (Khoury et al. MedRxiv 2021), suggesting that changing variants does not affect the relationship between neutralising titre and protection against infection. However we acknowledge this view is based on limited evidence and that further work in this area is needed and have discussed this in the limitations (page 14 line 21).

h) Lines 92-94: What do the authors consider as 'excellent' vaccine efficacy (VE)? Immune correlate analyses providing distinct IU50 nAb levels for different levels of VE were also performed for the Chadox-1 vaccine (Feng et al, Nat Med 2021) and the Ad26.COV2.S platform (Fong et al., Nat Microbiol 2022) and these should also be included in this section of the paragraph.

We considered 50% vaccine efficacy, as originally set as the threshold for approval by regulatory agencies such as the USA Food and Drug Administration (FDA) and European Medicines Agency (EMA), as high vaccine efficacy (VE). The VEs found in the referenced studies were 81-91% against symptomatic infection, which we consider excellent and we have added this to the text of the introduction. We thank the reviewer for highlighting the relevant study of the Ad26.COV2.S platform. We have referenced this in the introduction as this examined vaccine efficacy associated with an NT50 of 96.3 IU, which is very close to 100 IU. We have not referenced the analysis of the Chadox-1 vaccine at this section, as the only NT50 threshold examined in that study in international units was a threshold of 26 IU, which is considerably lower than the chosen 100 IU threshold. Feng et al also describe a live virus NT50 threshold, but this has not been normalised to an international standard and so is not discussed at this point. However this study is referenced in the discussion.

i) The authors repeatedly mention that the correlation between NT50 and RBD titers is weaker in the early post-infection period than in the later convalescent phase (e.g., lines 145-147 and lines 183-185). This is not entirely unexpected because (for example) IgM

antibodies will contribute to neutralization of serum early after infection (and less so late after infection) but will not be detected in the RBD assay that only identifies IgG antibodies.

We thank the reviewer for his important point, and have added this on page 5 line 10, and page 7 line 6.

Reviewer #3 (Remarks to the Author):

Thank you for the rebuttal. I have to say, the new version is improved.

However, I still have some concerns, which may not invalidate the findings of the study but suggest areas for further research and consideration.

1. The study primarily focuses on individuals who experienced acute symptoms of COVID-19 in the early stages of the pandemic or those who received vaccinations during a specific time period. However, there is a lack of long-term data on the dynamics of neutralizing antibodies and their correlation with RBD (receptor binding domain) titers.

We thank the reviewer for their comment. Compared to many of the vaccine trials discussed above, the current study includes individuals with a wide range of lengths of time post infection or vaccination (as outlined in table 1), with the aim of identifying a single threshold of binding antibody to maximise potential for clinical utility. We do examine the impact of time from the most recent booster and infection in the hybrid immunity group and find that time from the most recent infection, but not time from the most recent vaccine, impacts the association between RBD and BA.5 neutralising capacity.

However, despite this, we acknowledge that the cross sectional nature of this study limits the exploration of this relationship over prolonged periods of time and, specifically, how decay of RBD titres affects viral neutralisation capacity within specific individuals. We have now added discussion on this within the limitations (page 14 line 12).

As the reviewer rightly points out, there remain outstanding data gaps that are beyond the scope of the current manuscript that are areas of active research interest, including how a threshold of immunity can be used to predict the need for or optimal timing of booster

COVID19 vaccination. We consider that the analyses presented in this manuscript represent a significant step forward to facilitate further studies that explore the impact of various clinical factors on the utility of an IgG threshold in predicting protection against infection and disease. For example a recent nature medicine paper (Barnes et al. 2023) examined immune correlates of protection in those with different phenotypes of immunosuppression, finding an association between low RBD IgG titres and severe disease but arbitrarily using the lowest decile of RBD titres as the threshold for low titres. This manuscript provides a robust rationale for using a defined threshold titre of 456 BAU/ml in future similar analyses.

2. The study acknowledges the limitations of neutralizing assays, particularly live virus assays, due to their labor-intensive nature, high cost, and the requirement for high biosafety level facilities. These limitations hinder their use in large-scale vaccine trials or routine clinical settings. Additionally, there is significant inter-individual variation in the correlation between neutralizing antibodies and anti-spike IgG binding antibodies, which may impact the accuracy of predicting protective immunity solely based on binding antibody assays.

Please see below.

3. Furthermore, the study recognizes that the immune response to SARS-CoV-2 can vary depending on several factors, such as the individual's infection history, vaccine platform, number of vaccine doses, and time intervals between doses or previous infections. These factors contribute to the heterogeneity of population immunity, making it challenging to establish a universal threshold for protective immunity based solely on RBD titers.

We thank the reviewer for comments 2 and 3 and will address these points together. We agree that use of live virus neutralising assays, although considered the gold standard for measurement of viral neutralisation activity, are not feasible in clinical settings and that there is significant inter-individual correlation in potency, as discussed on page 3 line 17-23. We also acknowledge that there will be limitations to any defined IgG threshold (page 11 line 21).

However, our results add to a growing body of evidence suggesting antibody titres associate with a level of neutralising activity that is sufficient to protect against symptomatic COVID-19, and these could add value as part of a holistic clinical assessment (including individual risk factor profile, history of vaccine reactions, previous COVID-19 disease severity). We would hope this data will go some way towards enabling physicians to incorporate binding antibody

thresholds into clinical decision making. The current study uses the results of immune correlates analyses of early vaccine trials performed in WT and alpha dominant periods, updates them with data from prevalent VOCs and host immune profiles to provide a rationale for a defined threshold that remains robust across several waves of COVID19 that could be further evaluated in clinical trials.

4. The authors proposed a discrete RBD threshold as a predictor of protective neutralizing capacity against different variants of concern (VOC). However, it is essential to externally validate these findings in independent cohorts to ensure the reliability and generalizability of the proposed threshold.

We thank the reviewer for this important point. To improve the robustness of our findings we have now added data from an independent cohort. We utilised a cohort from the EU-COVAT-1 AGED study performed within the VACCELERATE network. In contrast to the other groups examined in this study, this cohort includes only individuals ≥ 75 years of age, and we used a commercially available serologic assay to ensure that our findings remain consistent with assays other than the in house assay used for the primary analyses. Within this external, independent study, we found that the 456 BAU/ml RBD IgG threshold once again accurately predicts loss of protective neutralising capacity against an immune escape variant (Beta variant) with a sensitivity of 92% (80-98%) a specificity of 96% (86-99%) and an accuracy of 94% (87-98%). We have included these additional data in the new section on page 10, line 5.

We do hope that these detailed responses adequately address the reviewer feedback.

REVIEWERS' COMMENTS

Reviewer #2 (Remarks to the Author):

I thank the authors for responding to my comments, most of which I feel have been adequately addressed.

I would like to make a few final notes to the responses to the following comments:

Comment b (on the correlation of live and pseudovirus neutralization assays):

I understand and appreciate the use of the WHO international standard to report on the outcomes of neutralization assays. These standards are particularly suitable for comparing results from different laboratories that measure activity (e.g., binding) against identical antigens using identical (or very similar) methods. Their use also is helpful for neutralization assays, although these biologically complex assays are much more difficult to harmonize even when using international standards. In addition to differences in the levels of antigen on viral particles or receptors/co-factors on target cell surfaces, they can also differ in the detection of different classes of neutralizing antibodies. For example, the monoclonal antibody S309 (sotrovimab) can potently neutralize in live virus assays but fails to do so in some pseudovirus assays (e.g., Fig. 2b in [10.1016/j.cell.2021.06.020](https://doi.org/10.1016/j.cell.2021.06.020)). If a serum sample with an assigned neutralizing activity of 250 IU/ml (international standard) were to be spiked with S309-like antibodies, it might subsequently be assigned a much higher result when tested in live virus assays than when tested in neutralization assays. This is why I feel that a correlation of the live virus assay used in this study with those used in the vaccine trials (lentivirus-based pseudovirus assay) would be informative and important. I do, however, acknowledge and appreciate that the authors now clearly state this potential limitation in the discussion.

Comment d (on testing neutralization to BA.5 in more than one cohort):

I understand the rationale provided for testing neutralization activity against the Beta variant. My main point is that the correlate analysis of RBD titers and BA.5 neutralization in the population of double vaccinated individuals would probably have highlighted the potential limitations of the RBD titer approach for predicting neutralization of highly divergent and immune evasive variants. As has been shown many times, two (ancestral) mRNA vaccine doses induce high levels of RBD antibodies but do not result in detectable or high levels of Omicron neutralization. Of course, many individuals will by now have some form of hybrid immunity or three or more antigen contacts through vaccination, so doing the

BA.5 analysis for the cohorts of individuals with hybrid immunity is certainly the most relevant (as was done by the authors).

Comment h (on immune correlate analyses following vaccine trials):

The paper by Feng et al. (Nat Med 2021; on Chadox-1) reports different IU50 values for different levels of vaccine efficacy (VE). It does not just include the result of 26 IU stated in the author's rebuttal letter, which is the level associated with 80% VE, but also a result of 140 IU/ml as a correlate for 90% VE.

Reviewer #3 (Remarks to the Author):

The authors addressed my concerns. No further comments.

REVIEWER COMMENTS

Reviewer #2 (Remarks to the Author):

I thank the authors for responding to my comments, most of which I feel have been adequately addressed.

We thank the reviewer for these supportive comments.

I would like to make a few final notes to the responses to the following comments:

Comment b (on the correlation of live and pseudovirus neutralization assays):

I understand and appreciate the use of the WHO international standard to report on the outcomes of neutralization assays. These standards are particularly suitable for comparing results from different laboratories that measure activity (e.g., binding) against identical antigens using identical (or very similar) methods. Their use also is helpful for neutralization assays, although these biologically complex assays are much more difficult to harmonize even when using international standards. In addition to differences in the levels of antigen on viral particles or receptors/co-factors on target cell surfaces, they can also differ in the detection of different classes of neutralizing antibodies. For example, the monoclonal antibody S309 (sotrovimab) can potently neutralize in live virus assays but fails to do so in some pseudovirus assays (e.g., Fig. 2b in 10.1016/j.cell.2021.06.020). If a serum sample with an assigned neutralizing activity of 250 IU/ml (international standard) were to be spiked with S309-like antibodies, it might subsequently be assigned a much higher result when tested in live virus assays than when tested in neutralization assays. This is why I feel that a correlation of the live virus assay used in this study with those used in the vaccine trials (lentivirus-based pseudovirus assay) would be informative and important. I do, however, acknowledge and appreciate that the authors now clearly state this potential limitation in the discussion.

We thank the reviewer for this further comment. We agree with the reviewer that this would be an interesting area for further study, but is beyond the scope of the current manuscript and is as such stated in the limitations (page 14 line 19-21).

Comment d (on testing neutralization to BA.5 in more than one cohort):

I understand the rationale provided for testing neutralization activity against the Beta variant. My main point is that the correlate analysis of RBD titers and BA.5 neutralization in the population of double vaccinated individuals would probably have highlighted the potential limitations of the RBD titer approach for predicting neutralization of highly divergent and immune evasive variants. As has been shown many times, two (ancestral) mRNA vaccine doses induce high levels of RBD antibodies but do not result in detectable or high levels of Omicron neutralization. Of course, many individuals will by now have some form of hybrid immunity or three or more antigen contacts through vaccination, so doing the BA.5 analysis for the cohorts of individuals with hybrid immunity is certainly the most relevant (as was done by the authors).

We thank the reviewer for the further comment. We agree that immunity induced by vaccines alone may not retain neutralisation against BA.5 but also recognise that most individuals now have hybrid immunity from (at times recurrent) infections and vaccinations and that was the rationale behind the choice of cohort for the analysis against BA.5. We also have avoided speculating on what the BA.5 neutralisation would have been in the absence of hybrid immunity, and have as such added this to the limitations (page 14 line 6-7).

Comment h (on immune correlate analyses following vaccine trials):

The paper by Feng et al. (Nat Med 2021; on Chadox-1) reports different IU50 values for different levels of vaccine efficacy (VE). It does not just include the result of 26 IU stated in the author's rebuttal letter, which is the level associated with 80% VE, but also a result of 140 IU/ml as a correlate for 90% VE.

We thank the reviewer for highlighting this data from the cited paper. However, as 140 IU is substantially higher than 100 IU, we have not included this citation on page 3, line 13 when discussing the data on the 100 IU cutoff studies but have referenced this important study both earlier in the introduction (page 3 line 7) and in the discussion (page 13 line 11).

Reviewer #3 (Remarks to the Author):

The authors addressed my concerns. No further comments.

We thank the reviewer for their supportive comments.

We do hope that these detailed responses adequately address the reviewer feedback.